# Antimicrobial Blue Light versus Pathogenic Bacteria: Mechanism, Application in the Food Industry, Hurdle Technologies and Potential Resistance

**DOI:** 10.3390/foods9121895

**Published:** 2020-12-18

**Authors:** Joshua Hadi, Shuyan Wu, Gale Brightwell

**Affiliations:** 1AgResearch Ltd., Hopkirk Research Institute, Cnr University and Library Road, Massey University, Palmerston North 4442, New Zealand; Joshua.Hadi@agresearch.co.nz (J.H.); Shuyan.Wu@agresearch.co.nz (S.W.); 2New Zealand Food Safety Science and Research Centre, Tennent Drive, Massey University, Palmerston North 4474, New Zealand

**Keywords:** antimicrobial blue light, pathogenic bacteria, food-borne bacteria, endogenous photosensitizers, porphyrins

## Abstract

Blue light primarily exhibits antimicrobial activity through the activation of endogenous photosensitizers, which leads to the formation of reactive oxygen species that attack components of bacterial cells. Current data show that blue light is innocuous on the skin, but may inflict photo-damage to the eyes. Laboratory measurements indicate that antimicrobial blue light has minimal effects on the sensorial and nutritional properties of foods, although future research using human panels is required to ascertain these findings. Food properties also affect the efficacy of antimicrobial blue light, with attenuation or enhancement of the bactericidal activity observed in the presence of absorptive materials (for example, proteins on meats) or photosensitizers (for example, riboflavin in milk), respectively. Blue light can also be coupled with other treatments, such as polyphenols, essential oils and organic acids. While complete resistance to blue light has not been reported, isolated evidence suggests that bacterial tolerance to blue light may occur over time, especially through gene mutations, although at a slower rate than antibiotic resistance. Future studies can aim at characterizing the amount and type of intracellular photosensitizers across bacterial species and at assessing the oxygen-independent mechanism of blue light—for example, the inactivation of spoilage bacteria in vacuum-packed meats.

## 1. Introduction

Annually, there are 600 million cases and 420,000 deaths associated with food-borne pathogens, with the majority of the disease burdens (550 million cases and 230,000 deaths yearly) attributed to diarrheal diseases [1]. Bacterial pathogenic agents are major contributors to these diarrheal infections, particularly *Salmonella enterica*, *Camplyobacter* spp. and *Escherichia coli* [1], and can linger in food-processing environments and food products (for example, minimally-processed foods, such as fresh-cut fruits and vegetables or raw seafood). These findings highlight the importance of robust sanitization systems in the food industry.

While heat is a potent germicidal agent, thermal processing of foods may lead to undesirable organoleptic properties and the loss of nutrients. Consumer perception of food safety has also been associated with aversion to chemical hazards, which include food preservatives, pesticides and drug residues [2,3]. Thus, there is a need for non-thermal sanitization technologies that are also free of chemicals.

The emerging non-thermal food-processing technologies include high-pressure processing (HPP) and pulsed electric field (PEF) [4,5,6]. However, the current forms of these technologies are more costly and less energy efficient—and thus less environmentally friendly—than thermal processing. For example, when used to pasteurize orange juice, HPP and PEF were estimated to consume 24–27 times more electricity (kW/year), incur 5–7 times higher total cost (cents/L) and emit 7–9 times more carbon dioxide than thermal pasteurization [7].

Alternatively, light-based technologies, particularly light-emitting diodes (LED), can be used as a cheap and sustainable non-thermal sanitization system [8]. It is known that ultraviolet-C (UV-C; particularly at 254 nm) exhibits bactericidal activities by inducing the formation of pyrimidine dimers in the bacterial genome and thus can be used within the food industry to sanitize food products or the processing environments [9,10]. However, health issues may arise from the use of ultraviolet (UV) radiation in the food industry, especially as constant exposure of workers to UV may increase the risks for skin cancer (for example, basal cell carcinoma, squamous cell carcinoma and malignant melanoma) [11,12]. A study also reported that accidental exposure of two healthcare workers to UV-C germicidal lamps (254 nm) led to bilateral keratoconjunctivitis and face erythema after 12–24 h, followed by other complications to the skin, eyes, nail and hair after 24 months [13].

In this review, we provide discussions on another emerging light-based sanitization technology derived from the blue region of the visible light spectrum, which is less detrimental to mammalian cells than UV [14] and thus allows for a wider application within the food supply chain due to its safety. We focus on studies that assessed the bactericidal efficacy of blue light-mediated technology on surfaces and in different food matrixes. Further, we also discuss the antimicrobial mechanisms of blue light, available technologies, safety aspects, the combination of blue light with other treatments (hurdle technology) and the potential development of bacterial tolerance to blue light. Additionally, a brief discussion on the inactivation of fish pathogenic bacteria (non-human pathogens) is also provided.

## 2. Pathogenic Bacteria in Food

Food-borne diseases are mainly caused by the consumption of contaminated food or water, with contamination possibly occurring at any point of production or distribution. Globally, the major food-borne pathogenic bacteria include *Salmonella* spp., *Campylobacter* spp., enterohemorrhagic *E. coli* (EHEC), *Listeria monocytogenes* and *Vibrio cholerae* [15]—the distribution of these bacteria across the globe, among other food-borne pathogenic agents, is summarized in a report by the World Health Organization [1].

In food-processing environments, Gram-negative bacteria are pre-dominant, particularly *Pseudomonas* spp., *Enterobacteriaceae* (especially *Serratia* spp.) and *Acinetobacter* spp. Among Gram-positive bacteria, lactic acid bacteria (LAB), *Staphylococcus* spp. and *Bacillus* spp. are the most commonly identified residential bacteria. While some of these tend to be innocuous, several pathogenic strains, such as *Staphylococcus aureus*, *Pseudomonas aeruginosa* and *Bacillus cereus*, are also known to linger on surfaces, especially due to their ability to form spores or biofilms [16].

The main pathogenic bacteria associated with dairy products are *L. monocytogenes*, *Salmonella* spp., *S. aureus, Cronobacter* spp. and Shiga toxin-producing *E. coli* (STEC) [17,18]. Dairy farm environments are a common habitat for *L. monocytogenes*, *Salmonella* spp. and STEC [19,20,21,22,23,24,25], whereas *S. aureus* is less prevalent in the environments and its transmissions are more likely to occur through contaminated animals (for example, those that have mastitis) [26,27]. There are no conclusive data on the natural environments of *Cronobacter* spp., especially *Cronobacter sakazakii* that is commonly associated with contaminated infant formulas [28,29], albeit these bacteria have been associated with plants [30,31,32] and animal feed [27,33]. The prevalence and type of dairy-associated pathogenic bacteria may also vary with animal source of the milk and geographical location [34,35]. Based upon these data, pathogens are mainly transferred to dairy products or processing environments from farm environments (soil, animal feed, etc.). For instance, two outbreaks (STEC O26:H11 in Italy and *L. monocytogenes* in Canada) were associated with contaminated dairy processing plants (cheese and milk plants) [36,37]—one study found that contaminated cheeses from a dairy plant had been distributed to approximately 300 retailers, which caused extensive cross-contamination [36]. Contamination at the retail level has also been reported, with *L. monocytogenes*, *Salmonella* spp., *Shigella* spp. and *E. coli* O157:H7 found in cheeses and raw milk [38,39,40].

Outbreaks related to farm-based meat products, such as beef and pork, are primarily caused by *Salmonella* spp. and EHEC (particularly *E. coli* O157:H7) [41,42,43]. Other bacteria have also been identified as causes of meat product-related outbreaks, namely *L. monocytogenes*, *S. aureus*, *Campylobacter* spp., *Clostridium* spp. and *B. cereus* [41,42,44]. While poultry products may also carry all of these pathogens, previous outbreaks were mostly caused by *Salmonella* spp., *Campylobacter* spp. or *Clostridium perfringens* [45,46,47,48,49,50,51]. In food-processing plants, fecal matters (for example, in hides or poultry skin) and aerosols generated during processing (for example, dehiding or evisceration processes) could facilitate the spread of pathogenic bacteria [44,52]. Several studies have also reported on the prevalence of pathogenic bacteria, including *E. coli* O157:H7, *Salmonella* spp., *Shigella* spp. and antibiotic-resistant *S. aureus*, in retail shops across countries—these bacteria were found on the products (raw or cooked beef, mutton, pork, chicken and turkey) and in the environments [39,53,54,55,56].

In seafood, the major pathogenic bacteria include *Vibrio* spp., *Salmonella* spp., *L. monocytogenes*, *Campylobacter* spp., EHEC, *Clostridium* spp. and *Shigella* spp., which could cause diseases ranging from mild gastroenteritis to life-threatening infections [57,58,59,60]. *Vibrio* spp. is ubiquitous in aquatic ecosystems, with infections in humans commonly associated with *Vibrio parahaemolyticus*, *Vibrio vulnificus* and *V. cholerae* [57,61,62]. Other bacteria, such as *Salmonella* spp. and *E. coli*, may also proliferate in bodies of water, particularly when contaminated with sewage effluents [57,63]. Cross-contamination during food production is the primary route of transmission for *L. monocytogenes* within the seafood industry and thus presents a major concern due to its ability to persist in the environment and to multiply during refrigeration [64]. Similarly, a study identified the contamination of a cutting board by *V. parahaemolyticus* from raw squid as a cause of a gastroenteritis outbreak at a food bazaar in South Korea [65]. Further, there is a heightened health risk in consuming raw seafood, as evident from previous outbreaks associated with uncooked (or undercooked) fish, oysters, abalone or sea squirt [66,67,68,69].

Fruits and vegetables may harbor a myriad of pathogenic bacteria, such as *Shigella* spp., *B. cereus*, *Campylobacter* spp., *Yersenia enterocolitica* and *Clostridium botulinum*, albeit previous outbreaks were mostly associated with STEC (particularly *E. coli* O157:H7), *Salmonella* spp. and *L. monocytogenes* [70,71,72]. Irrigation water that comes from contaminated sources is a major reservoir for these pathogens [73,74,75] and may occasionally carry *Vibrio* spp., for example, two studies identified *Vibrio* spp. on vegetables irrigated with untreated water from streams [76] and wastewater [77]. Other sources of contamination include pre-harvest factors, such as compost, insects, soil and wildlife animals, along with harvesting equipment or post-harvest vectors, including human (during packing), transport vehicles and processing equipment [70,71]. Pathogenic bacteria have also been detected in different horticultural products at the retail level across the globe: *L. monocytogenes* and *E. coli* isolated from frozen fruits or vegetables (England) [78]; *L. monocytogenes*, *S. enterica* or *E. coli* from ready-to-eat raw vegetables (UK, Malaysia or Nigeria) [79,80,81]; and *Salmonella enterica* subsp. *enterica* serovar Typhimurium, *C. perfringens*, *Campylobacter* spp. or *L. monocytogenes* from fresh produce (Mexico, Canada or New Zealand) [82,83,84]. In addition, a meta-analysis of 53 studies identified 453 cumulative incidences of STEC, *L. monocytogenes* and *Salmonella* spp. in fruits/vegetables from retail establishments across Europe between the years 2001 and 2017, with *L. monocytogenes* dominating in vegetables and STEC in fruits [85].

Bacteria may form biofilms to resist physical, mechanical and/or chemical stresses, including chemical disinfectants used in food-processing environments. For instance, several pathogenic staphylococci isolated from food or food equipment, namely *Staphylococcus capitis*, *Staphylococcus cohini*, *Staphylococcus saprophyticus* and *Staphylococcus epidermidis*, had shown abilities to form biofilms on polystyrene and stainless steel [86]. The stability of these biofilms against disinfectants (benzalkonium chloride) or denaturation enzymes (dispersin B, proteinase K or trypsin) is dictated by their compositions, which are determined by the presence of genes encoding either cell wall anchored proteins (CWA) or polysaccharide intracellular adhesin (PIA) [87]. This finding presents an alternative mechanism of biofilm formation in *Staphylococcus* spp., which is predominantly attributed to the presence of *ica* operon (*icaADBC* locus and *icaR* regulatory gene) that encodes PIA [88]—for example, *icaA* gene was found to be correlated to strong biofilm formation in food-related staphylococci isolates [86]. Other major biofilm-forming pathogenic bacteria include *L. monocytogenes* (poultry, red meat, seafood and dairy), *Salmonella* spp. (poultry, red meat, seafood and horticulture), *E. coli* O157:H7 (red meat and horticulture), *B. cereus* (dairy, seafood and horticulture), *Vibrio* spp. (seafood) and *Campylobacter* spp. (poultry) [89,90,91,92].

In addition, biofilms are composed of bacterial aggregates enclosed in extracellular polymeric matrix, which constitutes polysaccharides, proteins, lipids and exogenous deoxyribonucleic acids (DNA), and can function as a platform for physical/social interactions (for example, microbial consortia) that enhance gene transfers [93]. Several bacteria, such as *B. cereus* and *E. coli* O157:H7, also form multispecies biofilms to enhance their survival in food-processing lines [94,95]. The formation of biofilms is also dependent on bacterial structures that are responsible for initial surface attachment, such as flagella and/or fimbriae (for example, curli) in *L. monocytogenes* [96,97], *S.* Typhimurium [98,99], *E. coli* O157:H7 [100] and *V. cholerae* [101].

## 3. Antimicrobial Blue Light

### 3.1. Mechanism

Bactericidal effects of blue light are mostly attributed to the wavelength range of 400 to 450 nm [102], although several reports have demonstrated the antimicrobial efficacy of blue light at longer wavelengths (460, 465 or 470 nm) [103,104,105,106]. Blue light-mediated inactivation of bacteria is associated with the generation of reactive oxygen species (ROS) when the light is absorbed by endogenous photosensitizers, which can be found in different types of bacteria (Gram positive and Gram negative; aerobic and anaerobic) [107]. Given that these photosensitizers, such as protoporphyrin, coproporphyrin and uroporphyrin, are intermediate species in the heme biosynthesis, it is likely that they are accumulated in the cytoplasmic matrix [108,109], although their precise locations within the bacterial cell are not fully understood.

The blue light-mediated photosensitization process is dependent on the presence of oxygen and mainly induces cytotoxicity (apoptosis or necrosis) through oxidative stresses caused by singlet oxygen species (^1^O_2_) [110]. Upon illumination, photosensitizers at a ground state (lowest energy level) are converted into their excited singlet state (short-lived) or triplet state (long-lived), which, in the presence oxygen, can undergo two types of energy transfer: (1) type I that produces toxic oxygen species, such as hydrogen peroxide (H_2_O_2_), superoxide or hydroxyl radicals; (2) type II that generates ^1^O_2_ [111]. Subsequently, these ROS can induce damages to different parts of the bacterial cells, including the cell membrane, cell wall and genome (Figure 1).

An increase in blue light-induced membrane permeability was observed across several studies [112,113,114,115], although the precise mechanism is not fully elucidated. A study found that blue light illumination (405 nm) did not affect the lipid membrane of *Salmonella* spp.—there was an absence of malondialdehyde, which is a product of lipid peroxidation [116]. In contrast, two studies demonstrated that blue light inactivation (415 nm) of methicillin-resistant *S. aureus* (MRSA) or *C. sakazakii* involved lipid peroxidation, as determined by the detection of malondialdehyde and reduction in post-treatment unsaturated fatty acids (C_16:1_ in both bacteria, C_20:1_ and C_20:4_ in MRSA, and C_18:1_ and C_18:2_ in *C. sakazakii*) [113,115]. Further, while one study observed the presence of blue light-induced oxidation of guanine residues in the bacterial DNA of *Salmonella* spp. (presence of 8-hydroxydeoxyguanosine) [116], others reported no DNA breakage in blue light-treated (405 nm) *E. coli* O157:H7, *Shigella sonnei* and *S.* Typhimurium [112]. These discrepancies are potentially due to the fact that the type and amount of endogenous photosensitizers vary across different bacterial species, although further investigations are needed to explain the different susceptibilities of bacteria to blue light [117].

In addition to lipids and nucleic acids, blue light can also attack proteins, carbohydrates (polysaccharide) and peptidoglycan (polymers of amino acids and sugars in bacterial cell walls). Blue light treatments, in the presence of exogenous cationic photosensitizers, induced the loss of cell membrane-associated proteins in *S. aureus* [118] and the reduction of 81% in the polysaccharide content within *P. aeruginosa* biofilms [119]. In two studies, images taken by transmission electron microscopy revealed blue light-induced breakages of bacterial cell walls in MRSA [120] and *Acinetobacter baumannii* [121]. Further, *E. coli* lipopolysaccharide coated on titanium disc was inactivated upon illumination by blue light (405 nm), as evident from the reduced activities of mouse macrophages post-treatment [122]. However, the current literature lacks data on the effect of blue light on lipopolysaccharide (endotoxin) contained within intact outer membranes of Gram-negative bacteria and thus it is a subject of future studies.

Blue light can act as a transcriptional regulator in bacteria [127,128], especially due to the presence of blue light receptor proteins [128]. These photoreceptors include the blue light-sensing flavin adenine diphosphate (BLUF) proteins that can undergo conformational changes upon illumination by blue light and subsequently elicit downstream effects on bacterial surface attachments, biofilm formation and motility [129]. For instance, two studies found that a BLUF-associated protein, namely YcgF, downregulated the synthesis of curli fibres, but upregulated biofilm formation in *E. coli* [123,124]. In contrast, others reported that *A. baumannii*-harboring *blsA* gene, which encodes BLUF-containing photoreceptor proteins, did not form biofilms under blue light (462 nm), whereas biofilms were observed in a mutant strain with no functional *blsA* [125]. The viability of these bacteria was not affected by blue light in both wild and mutated strains, although blue light had a negative effect on bacterial motility and pellicle formation [125]. These findings indicate that there is a variety of blue light-sensing pathways in bacteria, which could be further explored as an alternative method for controlling the growth of bacteria. Another study also observed an alternative molecular mechanism of bactericidal blue light (460 nm) that involved the activation of prophage genes in MRSA, which subsequently led to the killing of the bacteria [126]. Future studies could aim at investigating the presence of similar genes (light-sensing and prophage genes) in food-borne bacteria and subsequently at designing targeted blue light-mediated interventions for controlling the persistence of these bacteria in food or food-processing environments.

In summary, antimicrobial blue light can act upon different parts of the bacterial cell, primarily through the action of ROS. These ROS can induce oxidative damages to a range of macromolecules, such as lipids (cell membrane), proteins (cell wall-associated proteins), nucleic acids (DNA, RNA or plasmids) and polysaccharides (extracellular matrix of biofilms). Additionally, several bacterial species, such as *E. coli* and *A. baumannii*, possess blue light receptors that control biofilm formation and motility, and thus can be targeted to reduce their persistence in the environments. Further, several prophage genes may be activated by blue light and induce inactivation of the carrying bacteria (Figure 1).

### 3.2. Available Technologies

The majority of studies on antimicrobial blue light have used light-emitting diodes (LED) as a light source. LED is commonly comprised of semiconductor materials that are doped with impurities, which create free electrons on the *n* side and holes (absence of electrons) on the *p* side—also known as the *p*–*n* junction. When electrical voltage is applied, current flows from the positively-charged end (*p* side; anode) to the negatively-charged end (*n* side; cathode), with electrons moving in the opposite direction. Subsequently, as an electron interacts with a hole, it falls to a lower energy state through the release of a photon. In this process, the resulting color emitted corresponds to the band gap energy within the *p*–*n* junction, which depends on the semiconductor materials and impurities used [130,131]. Currently, a typical blue LED is made of indium/gallium nitride (InGaN) layers grown on sapphire or silicon substrates, which can theoretically cover the entire visible light spectrum—365 nm (GaN) to 1771 nm (InN)—albeit the quality of materials deposited within the LED structure declines continuously beyond 480 nm due to a range of inherent material challenges [132].

Laser diode, which emits light with a higher coherence and narrower emission band than LED, is another source of blue light that has been used in clinical settings. The photomodulative effects of these two light sources on biological systems have been a subject of debates, especially due to their differences in light coherence and wavelength bandwidth. However, accumulating evidence suggests that these parameters have little effect on the biological efficacy of light-based technologies—for example, two studies found similar effects of red LED and laser diode upon tissue repair in rats [133,134]. Others also proposed that biological effects of light were dependent on dosage and wavelength, but not on light sources, with similar healing effects of LEDs and laser diodes on skin wounds reported across different studies [135]. Similarly, the antimicrobial potency of blue light is independent of the light source used, as one in vitro study revealed that LED (405 nm; non-coherent light) and laser diode (405 nm; coherent light) were equally efficient in inactivating MRSA across four light dosages (40, 54, 81 or 121 J/cm^2^) [136]. Although LED is relatively cheaper than laser diodes, it remains unclear which technology is more efficient based upon their germicidal output per unit electrical power input. In addition, superluminous diode (SLD; 405 or 470 nm), which is an intermediate between LED and laser diode in its light intensity, coherence and emission bandwidth [137,138,139], has also shown bactericidal activities against *P. aeruginosa*, MRSA and *S. aureus* in vitro [140,141,142].

A femtosecond laser, which emits ultrasecond pulses at approximately 10^−15^ s per pulse, is another technology that can be used to deliver antimicrobial blue light. At light dosages of 18.9–75.6 J/cm^2^ (5–20 min), a blue femtosecond laser (400, 410 and 420 nm) inhibited the growth of *S. aureus* and *P. aeruginosa* on agar plates (inhibition zones observed), possibly due to DNA damages induced by ROS [143]. In agreement, a femtosecond laser (425 nm; 800 J/cm^2^; 1 h) reduced a mutant *S.* Typhimurium lacking RecA proteins (reponsible for damaged DNA repair) by 5 log colony forming units (CFU), whereas only 0.5-log reduction (CFU) was observed for the wild-type bacteria and thus this finding enhanced the view that DNA damage was a predominant inactivation mechanism of a bactericidal femtosecond laser [144]. However, the two studies used different methods for measuring bactericidal activity (qualitative or quantitative), and also differed in their light dosages (max. 75.6 J/cm^2^ or 800 J/cm^2^) and treatment times (max. 20 min or 60 min) [143,144]. Therefore, the potential use of a femtosecond laser as an antimicrobial technology depends on future investigations into its energy efficiency and also its efficacy against different types of bacteria.

### 3.3. Blue Light Regimes

Antimicrobial blue light may be delivered at high irradiance with short duration times (HI-SD) or low irradiance with long duration times (LI-SD). A study demonstrated that the bactericidal activity of blue light (405 nm) was dependent on light dosage: the highest inactivation of four bacteria, namely *S. aureus*, *Streptococcus pneumoniae*, *E. coli* and *P. aeruginosa*, was achieved at the highest irradiance (approximately 9 mW/cm^2^) for a constant treatment time (120 min) or in the longest illumination time (250 min) at a constant irradiance (approximately 9 mW/cm^2^) [145].

In the same study, HI-SD treatment (approximately 9 mW/cm^2^ for 250 min) was also less effective than LI-SD (approximately 2.25 mW/cm^2^ for 1000 min) in inactivating pathogenic bacteria. Isolated colonies were observed on the perimeter of plates exposed to HI-SD, whereas confluent border present on LI-SD plates, indicating post-treatment migration of bacteria to the nutrient-rich and non-treated areas on HI-SD plates. Thus, LI-SD seemed to exhibit higher bactericidal and bacteriostatic effects on a qualitative level [145]. A similar finding was reported for *L. monocytogenes*, with LI-SD treatment of 10 mW/cm^2^ for 180 min yielding a 5.18-log reduction (CFU/mL), whereas HI-SD treatments of 20 mW/cm^2^ for 90 or 30 mW/cm^2^ for 60 min produced bacterial inactivation of approximately 5 log CFU/mL—the differences were not statistically significant [146]. However, neither study assessed the germicidal efficiency of HI-SD or LI-SD treatments per unit energy [145,146] and thus it remains inconclusive whether either regime is more suitable for practical applications in the food industry.

Alternatively, blue light can be delivered as pulses to increase its bactericidal efficiency. Pulsed blue light technology (450 nm; 33% duty cycle; three times a day for 3 days at 30 min intervals between each treatment) was reported to inactivate planktonic MRSA and *Propionibacterium acnes* (7 log CFU/mL) at light dosages of 7.6 and 5 J/cm^2^, respectively [147]. The same technology (7.6 J/cm^2^) also disrupted the biofilm networks of both bacteria and reduced the number of viable bacteria within the biofilm structures by approximately 1.89 and 1.56 log CFU/mL for MRSA and *P. acnes*, respectively [147]. In support of this view, pulsed blue LED (450 nm; 33% duty cycle) had a higher bactericidal efficiency against *P. acnes* than two other regimes (20% or 100% duty cycle), with a 7-log reduction (CFU/mL) achieved at a light dosage of 5 J/cm^2^ (2 mW/cm^2^ repeated nine times at 3-h intervals) [148].

For *S. aureus*, pulsed blue LED (405 nm; 25, 50 or 75% duty cycle) and continuous blue light (405 nm; 100% duty cycle) had similar inactivation efficiency (95–98%), albeit the pulsed blue light had approximately 83% higher optical efficiency (bacterial reduction in CFU/mL per J/cm^2^) [149]. Based upon these findings [147,148,149], pulsed blue light is preferred than continuous blue light in vitro, but its utilization in food settings is a subject of further investigations into its ability to remain energy efficient during scale up.

### 3.4. Safety of Blue Light

Safety assessments of blue light have mostly been conducted in clinical settings. Generally, exposure of skin to blue light is safe, albeit high fluences at certain wavelengths could induce cytotoxic effects. In one study, eight volunteers were exposed to blue light (380–480 nm; peak at 420 nm; 100 J/cm^2^ per day) for five consecutive days and the subsequent results of their skin biopsies were reported as follows: (1) no significant change in the expression of p53, i.e., no DNA damage; (2) no inflammatory cells and sunburn before and after treatment; (3) transient melanogenesis and vacuolization of keratinocytes observed, although these changes did not result in cell apoptosis [150]. Similarly, an in vitro study demonstrated that blue light (415 nm) could be used to inactivate *P. aeruginosa* on skin burns without inflicting any damage on the mouse skin at an effective antimicrobial dosage of 55.8 J/cm^2^ [151]. Exposure to the same blue light at a dosage of 109.9 J/cm^2^ inactivated human keratinocytes and *P. aeruginosa* by 0.16 log cell/mL and 7.48 log CFU/mL, respectively. However, cytotoxic effects of blue light on human endothelial and keratinocyte cells were observed at wavelengths of 412, 419 and 426 nm (66–100 J/cm^2^) or 453 nm (>500 J/cm^2^) [152].

In an in vitro study, damages on human corneal and conjuctival epithelial cells were observed after prolonged (17 h) exposure to blue light (420 and 430 nm at 1.13 and 1.16 W/cm^2^, respectively), with the authors reporting decreased cellular viabilities, morphological changes of the cells, accumulation of ROS and altered mRNA expression of biomarkers associated with cellular inflammatory response and antioxidant defense system [153]. A review article presented evidence of the adverse effects that blue light (415–455 nm) inflicted on retina (oxidative stress), lens (cataract due to accumulating ROS) and blood-retinal barrier functions [154]. Another group of researchers also reported the suppression of plasma melatonin in eight human subjects exposed to blue light (469 nm; corneal irradiance 0.1–600 W/cm^2^ for 90 min)—the extent of suppression was significantly higher at higher irradiances (*p* < 0.0001)—which suggests that blue light has the potential to disrupt circadian rhythm [155].

Widespread implementation of blue light-based technologies requires robust safety standards. According to the American Conference of Governmental Industrial Hygienists, daily exposure of workers to blue light is recommended to follow these rules: (1) for an exposure of 10,000 s (2.8 h) or more, the maximum intensity of the light source is ≤0.01 W/cm^2^.sr; (2) for light intensity above 0.01 W/cm^2^.sr, the maximum light dosage is 100 J/cm^2^.sr, where light dosage (J/cm^2^.sr) = light intensity (W/cm^2^.sr) × time of exposure (s); (3) for a light source subtending an angle less than 0.011 radian, the maximum light intensity is 10^−4^ W/cm^2^ for viewing durations greater than 100 s [156]. In accordance with these recommendations, a study analyzed blue light-related hazards through optical radiation measurements of several light sources [157]—the methodology in this study can be applied within the food industry for assessing the safety of different antimicrobial blue light technologies.

Based upon the findings presented in this section, blue light is innocuous on the skin, but deleterious to the eyes. Thus, safety glasses can be prescribed for personnel working within the proximity of high-intensity blue light sources. A study reported that several glasses and light filters significantly reduced (*p* < 0.001) the transmission of blue light from two LEDs (389–500 nm at 1625 mW/cm^2^ or 410–510 nm at 1680 mW/cm^2^; 10 s) by at least 97% [158]. Others reported that the use of blue light-blocking amber glasses improved the sleep quality of people with sleep disorders (self-reported or clinically diagnosed) [159,160], with an earlier endogenous dim-light melatonin onset observed when patients wore amber glasses [160]. However, there are no available data on whether anti-blue light glasses or filters can prevent damages to ocular cells. In addition, there is a need for a universal safety standard that governs the use of antimicrobial blue light within the food industry and thus the scientific community should aim at establishing the effective antimicrobial light dosages for different food-borne bacteria.

## 4. Application of Antimicrobial Blue Light on Surfaces and in Food Matrixes

### 4.1. Inactivation of Bacteria on Food Packaging and Work Surfaces

Heating effects induced by blue light treatment would be undesirable in industrial settings. Two studies reported that the surface temperatures of stainless steel increased to approximately 50–56 °C when treated with antimicrobial blue light (405 nm; 150–306 mW/cm^2^; 180–185 J/cm^2^) [161,162]. In these studies, bacterial reductions of 5 and <1 log CFU were achieved for *Campylobacter* spp. [162] and other pathogenic bacteria [161], respectively (Table 1). However, others observed a temperature increase of only 2.5 °C when stainless steel was continuously illuminated with blue light for 8 h (405 nm; 26 mW/cm^2^, 748.8 J/cm^2^) [163]. The discrepancies between these studies can be attributed to the different light intensities (mW/cm^2^) used and thus optimization studies are required to determine the suitable combination of light intensity and treatment time, i.e., high intensity-short duration or low intensity-long duration. For example, the reduction of blue light dosage from approximately 183–186 to 89–92 J/cm^2^ alleviated surface heating effects—final surface temperatures were approximately 44–56 and 31–36 °C at 183–186 and 89–92 J/cm^2^, respectively—although the bacterial inactivation was also reduced from 5 log CFU (183–186 J/cm^2^) to 1.1–3.1 log CFU (89–92 J/cm^2^) [162].

At 4, 15, and 20 °C, the formation of biofilm was inhibited by blue light (405 nm; 748.8 J/cm^2^) on stainless steel and acrylic coupons contaminated with *L. monocytogenes*-laden salmon exudates. However, the bacterial population within blue light-treated pre-formed biofilms was only significantly reduced (*p* < 0.05) at 25 °C [163]. This finding suggests that the blue light is more effective when used on cells contained in forming biofilms than in established biofilms. The efficacy of blue light in inactivating biofilms on other surfaces is a subject of future studies.

Blue light was able to traverse transparent solid surfaces, such as glass and acrylic slides, which was evident from the same inactivation rates of *E. coli* biofilms on top (direct exposure) or at the bottom (indirect exposure) of these slides, although four percent of the light irradiance was lost during transmission across both slides [164]. However, another study found that the inactivation of *L. monocytogenes* on tryptic soy agar was dependent on the ability of blue light (406–470 nm) to penetrate several packaging materials used to cover the agar—for example, no inhibition was observed when polyethylene + nylon was used, whereas maximum inhibition was obtained with polypropylene [165]. Thus, it is pertinent that packaging or surface materials are taken into considerations prior to designing blue light treatments intended to inactivate bacteria located behind these materials.

### 4.2. Inactivation of Bacteria in Dairy and Liquid Foods: Milk, Cheese and Orange Juice

Current data suggest that antimicrobial blue light is effective against pathogenic and spoilage bacteria in dairy and liquid foods. At least 3-log reduction was achieved in all studies reviewed (Table 2), with the extent of bactericidal efficacy depending on temperature and light wavelength. For milk products, two studies assessed the blue light-mediated inactivation of pathogenic bacteria in skim and whole milk, but no data are available on blue light inactivation in concentrated milk. A study reported that pulsed white light (200–1100 nm) was able to inactivate *E. coli* in skim and whole milk, albeit not in concentrated milk [170]. Thus, future research is needed to ascertain whether antimicrobial blue light can retain its bactericidal potency in milk products with varying total solid contents.

Interestingly, a study found that blue light inactivation of several bacterial strains in milk was more efficient than in a clear liquid matrix (PBS), except for *Mycobacterium fortuitum*. Two explanations were proposed: (1) blue light was absorbed by riboflavin (photosensitizer) in milk, as apparent from the significant (*p* < 0.05) reduction in the amount of riboflavin post-treatment, which subsequently generated ROS; (2) milk strongly scattered light and retained the light longer within its matrix, relative to PBS [171]. In contrast, blue light inactivation of *Campylobacter* spp. was significantly higher (*p* < 0.05) in transparent Brucella broth than in opaque chicken exudate [162]. These findings suggest that the type of solid particulate in liquid matrixes determines whether bactericidal efficacy of blue light is enhanced or attenuated.

Further, the bactericidal efficacy of blue light in liquid matrix also varies across different bacterial species/strains tested. In PBS, blue- light treatment (405 nm) resulted in a 5-log reduction (CFU/mL) of *C. jejuni* (18 J/cm^2^) [172] and *L. monocytogenes* (185 J/cm^2^) [173], whereas *Salmonella* spp. and *E. coli* O157:H7 was only reduced by less than 1.5 log CFU/mL at light dosages of 180–185 J/cm^2^ [172,173]. Thus, future studies are required to establish the effects of liquid opacity, particularly for liquid foods, on blue light inactivation of different bacteria.

Blue light inactivation of *E. coli* in liquid milk was more efficient at lower wavelengths and higher temperatures, with optimal treatment (reduction of 5 log CFU/mL and minimum color change) achieved at 405 nm, 13.8 °C and for 37.83 min [174]. This view was corroborated by others, who found that *S. enterica* in orange juice was inactivated by blue light (460 min; 4500 J; 92 W/cm^2^) to a higher degree at 12 or 20 °C than at 4 °C, although the inactivation rate was the same across these temperatures at higher light intensities (147.7 or 254.7 mW/cm^2^) [175].

Further, major milk components, namely proteins, lipids and lactose, were retained after 2 h of blue light treatment (720 J/cm^2^), albeit the loss of riboflavin (vitamin B_2_) had resulted in a bleaching effect that was perceptible to naked eyes [171]. In orange juice, blue light treatment also induced a color change in a temperature- and light intensity-dependent manner, particularly when the treatment was applied at a low intensity with long duration (light dosage was constant) [175].

On cheeses, blue light treatments were effective against *L. monocytogenes* and *Pseudomonas fluorescens* (Table 2), with no color changes observed in treated ricotta [176] and packaged slice cheeses [165]. Hyun and Lee (2020) also found that the efficacy of blue light on packaged sliced cheese was higher at 4 °C than at 25 °C [165].

### 4.3. Inactivation of Bacteria in Horticultural Products

Several studies found that blue light sanitization of fruits and vegetables was dependent on the type of product. Glueck et al. observed that the photosensitizer-mediated inactivation of blue light (435 nm; 33.8 J/cm^2^) was affected by the geometry of the food, with higher efficacy observed in flat-surfaced vegetables (cucumber, tomatoes and lettuce) than in those with complex structures (fenugreek seeds, mung bean seeds and mung bean germlings) (Table 3) [177]. In support of this view, three other studies showed varying bactericidal efficacies of blue light across different application media. Tortik et al. demonstrated that the combination of blue light (435 nm; 33.8 J/cm^2^) and curcumin (50 µM) reduced the bacterial load of *S. aureus* on peppers and cucumber by 2.5–2.6 log CFU [178], whereas an identical treatment in a clear liquid matrix (PBS) resulted in a 7-log reduction (CFU) in the number of *S. aureus* [179]. Buchovec et al. also found that *S.* Typhimurium was inactivated by chlorophyllin/chitosan-mediated blue light treatment (405 nm; 38 J/cm^2^) to a lower degree on strawberries (2.2 log CFU/mL) than in PBS (6.5 log CFU/mL) [180]. Possible explanations include the varying light-reflecting properties of different matrixes, the adsorption of photosensitizers onto the cuticle of vegetables/fruits and the presence of antioxidants in vegetables/fruits that reduced the efficacy of blue light [178,180].

In the majority of studies that we reviewed, blue light treatments resulted in no detrimental effects upon the sensorial and nutritional properties of fruits and vegetables. For instance, the antioxidant activities of cherry tomatoes, fresh-cut mangoes and papayas were retained after treatment with blue light (405 nm) [116,181,182]. Most nutrients (vitamin C, β-carotene and lycopene) were also preserved in blue light-treated papayas or fresh-cut mangoes (405 nm), although there was a significant increase (*p* < 0.05) in the amount of flavonoids during storage (4 °C or 20 °C) in the illuminated papayas [116]—flavonoid content remained stable in fresh-cut mangoes [182]. Others observed no adverse visual quality on blue light-treated strawberries (405 nm), as compared with the untreated controls [180]. In fresh-cut Fuji apples, polyphenol oxidase and peroxidase was inhibited by the combination of curcumin (2 µM) and blue light (420 nm) and thus there was significantly less browning (*p* < 0.01) in treated apples as compared with untreated controls [183]. On the contrary, blue light (460 nm) induced a bleaching effect in fresh-cut pineapples, as measured by the reduction in its yellowness index [184], although no human observers were used to determine whether this change would be perceived as undesirable.

The non-thermal nature of blue light treatments also allows for their application on low-moisture products, such as almonds, albeit improvements on its bactericidal efficiency would be required (at least 4-log reduction is needed) [185]. Additionally, blue light (405 nm) delayed the regrowth of *L. monocytogenes* on cherry tomatoes by 14 days, albeit this finding should prompt food producers to be vigilant in determining whether the blue light used is bacteriostatic or bactericidal against pathogenic bacteria [181].

### 4.4. Inactivation of Bacteria in Meat Products and Seafood (Chicken, Beef and Fish)

Generally, blue light treatment is less effective in meat and seafood products than on surfaces or in dairy and horticultural products (Table 1, Table 2, Table 3 and Table 4), possibly due to the presence of absorptive materials and ROS-neutralizing substances, such as proteins. For instance, the inactivation of *S.* Enteritidis by blue light was less efficient on cooked chicken meat (0.8–0.9 log CFU/cm^2^) than in the transparent PBS (1.3–2.4 log CFU/mL) [189]. However, blue light could still induce injuries on bacterial cells that render them more susceptible to subsequent stresses. On cooked chicken meat, *S.* Enteritidis lost its resistance to four antibiotics, relative to the untreated controls (details in Section 7.1.) [189]. Similarly, blue light treatment rendered *L. monocytogenes* and *Salmonella* spp. on fresh salmon significantly more susceptible (*p* < 0.05) to gastric digestion (pH 2) than untreated cells, especially at lower temperatures [190]. These findings indicate that bactericidal efficacy of blue light-mediated treatments could be improved by combining it with other treatments, such as organic acids, essential oils agents or polyphenols (details in Section 6).

The efficacy of blue light treatment on chicken products is dependent on the wavelength used, possibly related to wavelength-specific activation of different endogenous photosensitizers. Two studies found that *Campylobacter* spp. in chicken (fillet or skin) were reduced by 1.7–2.4 log CFU or 0.7–6.7 log CFU/g through treatments with blue light at 405 nm [162] or 395 nm [169], respectively. Other than the type of light, these differences may also be attributable to the different bacterial species, treatment distances and treatment lengths used (Table 4) [162,169]. In agreement, light dosage was found to be inversely proportional to treatment distance, for example, the inactivation of *C. jejuni* on chicken skin was higher at a treatment distance of 3 cm (6.7 log CFU/cm^2^) than at 12 cm (1 log CFU/cm^2^) or 23 cm (0.7 log CFU/cm^2^) (Table 4) [169].

Sensorial properties of chicken and salmon could be affected by light treatment, especially when using light at the UV–vis region or photosensitizers. A study reported on the heating effects of ultraviolet/blue light (395 nm), especially at shorter treatment distance and longer treatment length, which resulted in significant color changes (*p* < 0.05) in chicken fillet and skin [169]. Others found that while discoloration was absent in smoked salmon illuminated with blue light alone (460 nm), the whiteness index significantly increased (*p* < 0.05) in samples treated with riboflavin-mediated blue light, relative to the untreated control samples [191]. Consistently, an extended illumination (8 h) of fresh salmon with blue light alone (405 nm) did not result in color changes [190].

Further, the introduction of exogenous photosensitizers could improve the inactivation of pathogenic bacteria on cooked food. The combination of curcumin (50 or 100 µM) and blue light (435 nm; 33.8 J/cm^2^) resulted in a 1.7-log reduction (CFU) of *S. aureus* in cooked chicken meat, whereas the treatment of blue light alone had no effect on the bacterial load. Albeit, the authors suggested that the lipophilicity of curcumin could make it susceptible to attenuation by the fatty regions on the chicken skin and thus modification of this photosensitizer (or alternative photosensitizers) would be required to achieve a higher bactericidal activity [178]. Another study found that blue light (460 nm) reduced the population of *L. monocytogenes* on smoked salmon fillets by up to 1.12 log from the initial concentration of 3.5 log CFU/cm^2^, but only when riboflavin was present. The light dosage required to achieve the first log reduction was lower at 4 °C (1600 J/cm^2^) than at 12 °C (2000 J/cm^2^), although the difference was not statistically significant [191].

## 5. Potential Application of Blue Light in Food Supply Chain

### 5.1. Food Processing and Farms: Airborne and Surface Inactivation

Practical applications of any blue light technology within the food industry are dependent on its ability to inactivate pathogens over distances beyond those typically used in laboratory-scale experiments. In clinical settings, three studies found that a ceiling-mounted high-intensity narrow-spectrum light environmental decontamination system (HINS-light EDS; 405 nm) significantly reduced (*p* < 0.05) the total viable counts, including MRSA and *S. aureus*, on surfaces (for example, bed, table, chair, worktop or bins) [193,194,195]. The safety of HINS-light EDS also allowed it to be operated in the presence of humans, such as patients and healthcare workers, which is in contrast to ultraviolet germicidal lamps [193,194,195].

These findings suggest that there is a potential for HINS-light EDS (or similar technologies) to be used for environmental sanitization in food-processing plants. While blue light inactivation of planktonic and biofilm-associated bacteria has been tested on food packaging and also on work surfaces (for example, stainless steel, acrylic or glasses), there are limited studies on the sporicidal effects of blue light on food packaging (Table 1). In suspensions, antimicrobial blue light (405 nm; 1730 J/cm^2^) also reduced the population of bacterial endospores, namely those of *B. cereus*, *B. subtilis*, *Bacillus megaterium* and *Clostridium difficile*, by 4 log CFU/mL [196]. Thus, future in vivo validations are required to assess the ability of blue light to inactivate different forms of bacteria on a range of surfaces and at varying distances.

Further, bacteria may be aerosolized during food processing, persist in the air and subsequently spread across indoor premises. Interestingly, a study found that aerosolized *S. epidermidis* was significantly inactivated by approx. 2 log CFU/mL (*p* < 0.001) by blue light (405 nm; 39.5 J/cm^2^), with the susceptibility of the bacteria to blue light being 2–4 times higher in aerosols than in liquids or on surfaces [197]. Future studies should explore the potential of blue light for inactivating food-borne bacteria in aerosols.

Light treatments could also be used to treat veterinary diseases in farm animals, such as mastitis in cows. For example, there were significantly lower (*p* < 0.05) bacterial loads of *Streptococcus dysgalactiae* and coagulase-negative staphylococci in milk produced by cows treated with the combination red LED (635 nm; 200 J/cm^2^) and toluidine blue (2%), relative to the untreated groups—this treatment was not intended for direct decontamination milk, but for alleviating incidences of mastitis in cows, with bacteriological characteristic of milk samples only used as an indicator [198]. In vitro, bacteria isolated from bovine mastitis, namely *S. dysgalactiae*, *S. aureus* and *Streptococcus agalactiae*, were inactivated by red LED (662 nm; 3–12 J/cm^2^) and methylene blue (50 µM) [199]. Currently, there are no data available on the application of blue light against farm-animal pathogens. However, given the fact that blue light can act on endogenous chromophores, it may have practical advantages over the existing red LED that depends on exogenous photosensitizers to inactivate bacteria on cows.

### 5.2. Aquaculture

Pathogenic bacteria that attack fish include *Vibrio* spp., *Photobacterium damselae* subsp. *piscicida*, *Edwardsiella tarda* and *Edwardsiella ictaluri* [200]. Although most of these bacteria are not known to infect humans, high incidences of disease in farmed fish may inflict adverse economic consequences upon fish farmers. Thus, the availability of methods for inactivating these bacteria in aquaculture systems is paramount to sustain viable fisheries. Several studies have assessed the application of antimicrobial blue light against fish pathogens. In PBS, the blue light inactivation of several pathogenic fish bacteria was 132–543.7 and 247–2178 J/cm^2^ at 405 and 465 nm, respectively. Generally, these bacteria were more susceptible to blue light at 405 nm than at 465 nm, although there were variations across different bacterial species (Figure 2) [201].

The presence of particulates in aquaculture water reduced the bactericidal efficacy of artificial white light (380–700 nm), in combination with a cationic porphyrin (Tri-Py^+^-Me-PF), against *Vibrio fischeri*: (1) in unfiltered water, 50 µM of porphyrin and 43.2–64.8 J/cm^2^ of light were required to achieve a 7-log reduction (CFU/mL); (2) in filtered water, the combination of porphyrin (at least 10 µM) and light (64.8 J/cm^2^) led to a bacterial reduction of 7 log CFU/mL. When tested in PBS, a complete inactivation of *V. fischeri* (7 log CFU/mL) was achieved, regardless of variations in acidity (pH of 6.5–8.5), salinity (20–40 g/L), temperature (10–25 °C) and oxygen concentration (5.3–5.9 mg/L), although the rate of inactivation was highest at the physiological pH (7.4) and ambient temperature (25 °C) [202]. However, when similar treatments (white light at 380–700 nm; Tri-Py^+^-Me-PF at 5–50 µM) were used against heterotrophic bacteria cultivated from aquaculture water samples, the bactericidal efficacies varied across different water samples, ranging from 1.2 to 2 log CFU/mL. Nevertheless, in PBS, a complete inactivation (8 log CFU/mL) was observed for several pathogenic bacteria isolated from aquaculture water, namely *Vibrio* spp., *P*. *damselae*, *Enterococcus faecalis*, *E. coli* and *S. aureus*, after exposure to a combined treatment of artificial white light (380–700 nm; up to 648 J/cm^2^) and Tri-Py^+^-Me-PF (5 µM) [203].

In the absence of photosensitizer, blue LED (405 or 465 nm) was able to significantly reduce (*p* < 0.05 or 0.01) the bacterial loads of *Edwardsiella piscida* in rearing water, which subsequently also decreased the number of bacterial infections in Fancy carps (*Cyprinus caprio*) [204]. The light treatment did not induce damages on the fish eyes and skin, with also no increase in the production of heat-shock proteins or unusual feeding behaviour observed in the treated fish, relative to the untreated controls [204]. Albeit, precautions are needed as continuous exposure of some fish, such as sea bass and sole, to blue light (435–500 nm) could result in increased malformations and poor survival of the fish larvae [205]. Thus, additional studies are needed to assess the effects of antimicrobial blue light on live seafood, including fish, oysters and mussels.

### 5.3. Retail: Prolonging Shelf-Life

As previously discussed, food contamination at retail establishments could lead to outbreaks (Section 2), particularly as inactivation treatments are not usually present at this stage within the food supply chain. In Section 4, we have reviewed studies on the use of antimicrobial blue light against pathogenic bacteria in an array of food products and also on surfaces. However, the use of blue light at the retail level also requires it to inhibit spoilage microorganisms and thus extends the shelf-life of foods. There are also concerns about spoilage bacteria surviving cleaning regimes and persist on surfaces in food-processing plants, including *Pseudomonas* spp., *Serratia* spp., *Hafnia* spp. and LAB [16], and thus additional control measures are required. Further, several review articles have identified major spoilage microorganisms in dairy [206,207], horticultural [208], meat [209] and seafood [210] products, which may be a subject of future studies on antimicrobial blue light.

Although still limited in number, there are several studies that assessed blue light-mediated inactivation of spoilage microorganisms in food products and subsequently the shelf-life of these treated foods. For example, blue light significantly reduced (*p* < 0.05) the initial loads of mesophilic bacteria, yeasts or other microfungi on strawberries and cherry tomatoes, which delayed the spoilage onset by 2 and 4 days, respectively (Table 3) [181,188]. In Hami melon (cantaloupe), the combination of curcumin (50 µM) and blue LED (470 nm) significantly reduced (*p* < 0.05) the initial amount of total aerobic microorganisms by 1.38 log CFU/g, with the treated melons also having 1.8-log lower bacterial load (CFU/g) than the untreated controls after 9 days of storage at 4 °C. The soluble solid content, color, water content and firmness of Hami melon were also better preserved in the blue light-treated group than the untreated controls [211].

The combination of curcumin (10 µM) and blue light (470 nm; 5.4 J/cm^2^) extended the shelf-life of fresh oysters from 8 to 12 days at 4 °C, as determined by total aerobic plate count (shelf-life limit of 10^7^ CFU/g) and total volatile basic nitrogen analysis (shelf-life limit of 30 mg *n*/100 g oyster) [212]. Sensorial properties of the treated oyster were also improved at the shelf-life terminal point of the untreated group (8th day): (1) human panels rated treated oyster more favorably than untreated control in terms of smell, body color, mucus appearance and texture; (2) electronic nose indicated that the generation of spoilage metabolites was reduced in the treated group, relative to untreated controls. Retention of flavorful free amino acids and reduction in the oxidation of lipids and fatty acids were also observed in the blue light-treated group [212]. Similarly, the treatment of fresh sturgeons with curcumin (30 µM) and blue light (470 nm) significantly reduced (*p* < 0.05) the prevalence of spoilage *Pseudomonas* spp. during storage at 4 °C for 6–9 days [213].

The number of spoilage *P. fluorescens* was lowered by 1.85–3.60 log CFU/g in blue light-treated (460–470 nm) packaged sliced cheese, relative to untreated controls, after 2 and 7 d of storage at 25 and 4 °C, respectively (Table 2) [165].

## 6. Hurdle Technology

### 6.1. Photosensitizers

In all studies that combined exogenous photosensitizers with blue light treatments, the addition of photosensitizing agents improved the bactericidal efficacy of blue light, relative to when blue light was used alone (Table 1, Table 3 and Table 4). These photosensitizers facilitate the production of ROS, which subsequently induces bacterial inactivation. However, the bactericidal efficacy may also vary with the type of photosensitizer used. For instance, the combination of blue LED (427–470 nm; 30 J/cm^2^) and rose bengal (160 µg/mL) inactivated 7.1 log CFU/mL of *Porphyromonas gingivalis*, whereas when the same blue light was combined with erythrosine or phloxine, the bacterial reductions were only 0.9 or 1 log (CFU/mL), respectively [214].

Similarly, the efficacy of photosensitizers also depends on the delivery method within different application matrixes. For example, curcumin bound to polyvinylpyrrolidone did not improve the blue light inactivation on chicken skin, but was effective on vegetables. The lipophilic curcumin may be readily released by the hydrophilic polyvinylpyrrolidone to the fatty regions of the chicken skin. In contrast, micellar formulation of curcumin (NovaSol^®^-C) was effective on chicken skin, potentially due to the retention of curcumin in micellar form prior to contact with the bacterial cells [178].

Non-toxic inorganic salts can be used to potentiate photosensitization in photodynamic treatments, particularly through the production of non-oxygen reactive species, such as azide radicals from sodium azide salts or reactive iodine species from potassium iodide [215]. In an in vitro study, the combination of blue light (415 nm; 10 J/cm^2^), Photofrin (10 µM) and potassium iodide (100 mM) inactivated 6 log of five Gram-negative bacterial species, namely *E. coli*, *P. aeruginosa*, *Klebsiella pneumoniae*, *Proteus mirabilis* and *A. baumannii*, whereas in the absence of potassium iodide, photosensitization treatments resulted in no inactivation [216].

Precursors of endogenous photosensitizers, such as 5-aminolevulinic acid (ALA), can also be added to facilitate photo-inactivation of bacteria [110]. A study found that ALA and its derivatives induced the formation of photo-active porphyrins in Gram-negative (*E. coli* K-12, *E. coli* Ti05 and *P. aeruginosa)* and Gram-positive (*S. aureus*) bacteria, although the amount or type of porphyrin produced, and also the extent of bacterial photo-inactivation (white light; 120 J/cm^2^), depended on three factors: (1) type of precursor used; (2) bacterial species/strain tested; (3) concentration of precursor added [217]. In support of this view, another study demonstrated that ALA-mediated inactivation of bacteria by blue light (407–420 nm; 50–100 mJ/cm^2^) was more profound in Gram-positive (5–7 log CFU/mL; except for *B. cereus* and *Streptococcus faecalis*) than in Gram-negative bacteria (1–2 log CFU/mL)—this difference was possibly due to the higher amount of coproporphyrin present in Gram-positive bacteria tested (*B. cereus* produced 37–45% lower coproporphyrin than the other Gram-positive bacteria and was reduced by only 1–2 log CFU/mL; porphyrin production was not observed in *S. faecalis* and no reduction was reported) [218].

### 6.2. Acidity and Temperature

In clear liquid suspensions, the susceptibility of *E. coli* and *L. monocytogenes* to blue light (405 nm) was enhanced in the presence of environmental stresses: (1) the light dosages required to inactivate both bacteria (5 log CFU/mL) at stressful temperatures (4 °C or 45 °C) were significantly lower (*p* < 0.05) than at a non-stressful temperature (22 °C); (2) the light dosages required to inactivate *E. coli* and *L. monocytogenes* at pH 3 were reduced by 77% and 50%, respectively, relative to that required at pH 7; (3) the bacterial inactivation was significantly higher (*p* < 0.05) under osmotically-stressful conditions (salt concentrations of 10% and 15% for *E. coli* or 10% for *L. monocytogenes*) than under non-stressful conditions (salt concentrations of 0 or 0.8%) [219]. On nitrocellulose surface, both bacteria were also inactivated by blue light (405 nm; 36 J/cm^2^) to a higher extent at pH 3 (reduction of 95–99% reduction) than at pH 7 (reduction of 13–26%) [219]. In addition, the type of acid present determined the extent of bactericidal inactivation of blue LED (461 nm; 596.7 J/cm^2^) against *E. coli* O157:H7, *S.* Typhimurium, *L. monocytogenes* and *S. aureus*, with the highest inactivation rates at pH 4.5 achieved using lactic acid, followed by citric and malic acids [220].

However, there has been no consensus on how temperature affects the efficacy of antimicrobial blue light, with current data suggesting that it depends on the bacterial species and light wavelength used. The growth of *S.* Enteritidis on cooked chicken meat was only delayed when treated with blue light (405 nm) at 10 and 20 °C, but it was inactivated at 4 °C [189]. Similarly, five *S. enterica* serovars on fresh-cut pineapple, namely Typhimurium, Newport, Gaminara, Montevideo and Saintpaul, were inactivated by blue light (460 nm) at 7 and 16 °C (bactericidal), but only inhibited at 25 °C (bacteriostatic) [184]. On the contrary, *S. aureus*, *Lactobacillus plantarum* and *V. parahaemolyticus* in PBS were inactivated by blue light (405 or 460 nm) at all experimental temperatures (4, 10 or 25 °C), albeit the extent of bactericidal effect varied across bacterial species and also light wavelengths [221]. Another study also demonstrated that *E. coli* O157:H7, *L. monocytogenes* and *Salmonella* spp. on fresh-cut mangoes were inactivated by blue light (405 nm) at 4 and 10 °C, but only inhibited at 20 °C, with both bactericidal and bacteriostatic effects varying with bacterial species [182]. Interestingly, the population of *L. monocytogenes* in pre-biofilms on stainless steel and acrylic coupons was significantly reduced (*p* < 0.05) at 25 °C, but not at 4 °C [163].

### 6.3. Nanoparticle

Silver nanoparticles (AgNPs) are able to interact with negatively-charged molecules in bacterial cells, such as proteins or nucleic acids and subsequently induce damages to the cells, for example, by increasing cell membrane permeability, causing DNA damage or inhibiting protein synthesis [222]. A review by Carbone et al. summarized available data on the potential use of AgNPs as an antimicrobial agent in food packaging [223]. In photodynamic treatments, the combination of blue light (460 nm) with AgNPs was significantly more effective (*p* < 0.001) against MRSA and *P. aeruginosa* than each treatment alone [224,225]. The formation of *P. aeruginosa* biofilm on gelatin-based discs was also significantly inhibited (*p* < 0.001) by the combined treatment, relative to treatments with blue light or AgNPs alone [225].

Similarly, metal oxides can be used in photocatalytic processes to generate superoxides or hydroxyl radicals for inactivating microorganisms or oxidizing organic substances. In an in vitro study, zinc oxide nanoparticles (ZnO-NPs; 0.5 mg/mL) were combined with blue light (462 nm; 5.4 J/cm^2^) to inactivate 5 log CFU/mL of *A. baumannii*, with ZnO-NPs and blue light alone resulted in zero and less than 1-log reduction (CFU/mL), respectively. The combination of ZnO-NPs (0.1 mg/mL) and blue light (462 nm; 10.8 J/cm^2^) also significantly reduced the number of antibiotic-resistant (colistin or imipenem) *A. baumannii* (*p* < 0.005), *K. pneumoniae* (*p* < 0.005) or *Candida albicans* (fungus; *p* < 0.05) in culture medium. Further, transmission electron microscopical image revealed that the cell membrane was damaged in *A. baumannii*, but an analysis using gel electrophoresis showed no fragmentations of plasmid DNA post-treatment and thus these findings indicate that ZnO-NPs/blue light photocatalysis only attacks cytoplasmic membrane and not the bacterial genome [226].

### 6.4. Plant Extracts: Polyphenols and Essential Oils

Polyphenols are known to exhibit antimicrobial potency through their ability to bind to the cell membrane, cell wall and their associated proteins and thus compromising the structural integrity of the bacterial cell [227,228]. Inhibition of bacterial adhesins and quorum sensing by polyphenols also resulted in failures to form biofilms [228]. Interestingly, several studies have reported on the synergistic antimicrobial effects of blue light and plant-derived polyphenols, particularly at the blue light wavelength range of 385–400 nm. For example, gallic acid (4 mM) was combined with blue light (400 nm; 72 J/cm^2^) to inactivate 7.5 log CFU/mL of *S. aureus*, with lipid peroxidation observed through the detection of malondialdehyde. This lipid peroxidation was likely to be caused by the formation of hydroxyl radicals, which were detected by electron spin resonance [229]. In a follow-up study, the combination of blue light (400 nm; 75–150 J/cm^2^) and several polyphenols (1 mg/mL; caffeic acid, gallic acid, chlorogenic acid, epigallocatechin, epigallocatechin gallate and proanthocyanidin) induced significant inactivation (*p* < 0.05 or *p* < 0.01) of *E. faecalis*, *S. aureus*, *Streptococcus mutans*, *Aggregatibacter actinomycetemcomitans*, *E. coli* and *P. aeruginosa*. Damage to the DNA was also reported, which suggested that the polyphenols were incorporated into the bacterial cells, probably facilitated by the high affinity of polyphenols to the cell membrane [230].

Similarly, inactivation of *S. mutans* within biofilms was reported after exposure to the combination of caffeic acid (0–2 mg/mL) and light (365, 385 and 400 nm; 120–480 J/cm^2^), with the highest inactivation (5 log CFU/mL) achieved at caffeic concentration of 2 mg/mL and blue light dosage of 480 J/cm^2^ (385 nm) [231]. Other authors also reported the synergistic antimicrobial activities of blue light (400 nm) and wine grape-derived polyphenols (for example, catechin and its isotopic ingredients) against *S. aureus* or *P. aeruginosa* (5 log CFU/mL) [232,233].

In addition, essential oils possess antimicrobial activity by inducing membrane breakage and permeability, albeit Gram-negative bacteria are known to be more resistant than Gram-positive bacteria due to the presence of hydrophilic outer membrane [234,235]. Membrane leakages can lead to loss of cellular constituents, such as ions, genetic materials or adenosine triphosphate (ATP), and subsequently cell death [236,237]. One study found that essential oils derived from eucalyptus (5%), clove (0.5%) and thyme (0.5%) improved the blue light (469–470 nm) inactivation of *S. epidermidis* and *P. aeruginosa* by 2–7 and 3–8 log CFU/mL, respectively, as compared with inactivation achieved by light alone. Relative to treatments with essential oils alone, samples treated with the combination of essential oils and blue light had 3–6 and 3–5 log CFU/mL lower counts of *S. epidermidis* and *P. aeruginosa*, respectively [238].

## 7. Blue Light versus Antimicrobial Resistance and Consequences of Sub-Lethal Light Exposures

Antimicrobial resistance presents a challenge to the food industry, with multiroute transmissions of resistant bacteria occurring through the contamination of food-processing environments, transfer of genes originating from microorganisms intentionally added to foods (starter cultures, bio-preservative bacteria or bacteriophage) and cross-contamination of foods [239,240,241,242]. Throughout the years, antimicrobial-resistant pathogenic bacteria have emerged across different food industries: horticultural (vegetables and fruits) [240], seafood [241] and meat (food and livestock) [242]. In previous sections, we have reviewed several studies that successfully used blue light against drug-resistant bacteria, such as MRSA. However, in this section, we focus on the ability of blue light to sensitize multidrug-resistant bacteria to antibiotics and subsequently on the inactivation of biofilms (monomicrobial or polymicrobial) and also on the potential development of bacterial tolerance/resistance to blue light.

### 7.1. Resistant Bacteria: Improved Sensitivity to Antibiotics

Generally, bacteria achieve antimicrobial resistance through three mechanisms: (1) preventing the drug from reaching the target (limiting uptake or active efflux); (2) modifying the target sites, such as alterations of penicillin-binding proteins in Gram-positive bacteria; (3) inactivating the drug through degradation or chemical modulation [243]. Theoretically, blue light could induce damages that disrupt the ability of bacteria to limit drug uptakes or to perform active efflux (mechanism 1). As mentioned previously (Section 3.1), blue light could induce breakage of cell walls, increase the permeability of the cell membrane (lipid peroxidation) and inactivate lipopolysaccharides (constituents of outer membrane in Gram-negative bacteria)—all of these structures play roles in conferring barriers against antibiotics [243]. In addition, one study showed that blue light-treated MRSA suffered from potassium ion leakages, which suggested that several transmembrane proteins (for example, Na^+^/K^+^ ion pumps) may have been denatured [113] and thus potentially limiting the role of these transmembrane proteins in pumping drugs out of the bacterial cells (active efflux) [243].

Several studies also found that blue light rendered bacteria more susceptible to antibiotics. In an in vitro study, blue light-treated (411 nm; 150 J/cm^2^ per cycle; 15 cycles) *S. aureus* (methicillin-sensitive and resistant) had a higher susceptibility to gentamycin and doxycycline, but not vancomycin, ciprofloxacin, chloramphenicol and rifampicin, than untreated controls [244]. Another report showed that the minimum inhibitory concentration (MIC) of gentamycin, ceftazidime and meropenem against drug-resistant *P. aeruginosa* was reduced by up to 8-fold in the blue light-treated groups (405 nm; 10 or 12 J/cm^2^), relative to the untreated controls [245]. On cooked chicken meat, blue light-treated (405 nm; 1700 J/cm^2^) *S.* Enteritidis became more susceptible to ampicillin, chloramphenicol, nalidixic acid and rifampicin, relative to freshly-cultured or non-illuminated controls [189]. On the contrary, one study reported no change in the susceptibility of *S. aureus* (methicillin sensitive and resistant) to antibiotics (panel of 10, including gentamycin) after fifteen cycles of sub-lethal exposures to blue light (405 nm; 108 J/cm^2^ per cycle) [246].

Further, He et al. found that blue light could be combined with tetracycline-class antibiotics to inactivate drug-resistant *E. coli* and MRSA: (1) demeclocycline (DMCT; 10–50 µM) could be activated as a photosensitizer by blue light (415 nm; 10 J/cm^2^) and reduced the bacterial load of resistant *E. coli* and MRSA by 6 log CFU/mL; (2) sub-lethally injured *E. coli* underwent inactivation during the sub-culturing of these bacteria post-treatment. This indicated that the antibiotic was still active, even in the absence of light, and continued to inactivate the sensitized bacteria by inhibiting their ribosomes; (3) minimum inhibitory concentrations of DMCT, doxycycline, minocycline and tetracycline against drug-resistant *E. coli* and MRSA were reduced by up to 8-fold, when applied in the presence of blue light (415 nm), relative to the dark controls [247].

### 7.2. Inactivation of Biofilms

Biofilms facilitate horizontal gene transfers between individual bacteria within the matrix, especially in those that contain more than one bacterial species. These exchanges of mobile genetic elements may lead to an increase in antimicrobial resistance and environmental persistence [248]. To mitigate this issue, a group of researchers deployed antimicrobial blue light (405 nm; 500 J/cm^2^) against polymicrobial biofilms and achieved log reductions of 2.37 and 3.40 CFU/mL for MRSA and *P. aeruginosa*, respectively, within a dual-species biofilm. The same blue light treatment on another dual-species biofilm inactivated *P. aeruginosa* and *C. albicans* by 6.34 and 3.11 log CFU/mL, respectively. As expected, monomicrobial biofilms were more susceptible to blue light, with damages to the exopolysaccharide matrix also observed across different types of biofilm [249].

The efficacy of blue light against biofilms also varies across bacterial species. For example, blue light (405 nm; 108–206 J/cm^2^) significantly inactivated monomicrobial biofilms of drug-resistant *A. baumannii*, *P. aeruginosa* and *Neisseria gonorrhoeae* (4–8 log CFU/mL; *p* < 0.01 or *p* < 0.0001), whereas the same blue light treatment did not significantly affect the biofilms of *E. coli*, *E. faecalis* and *Proteus mirabilis*. Further, blue light (405 nm; 216 J/cm^2^) significantly reduced (*p* < 0.01) the number of MRSA in biofilms grown for 24 h, but not in biofilms grown for 48 h [250].

### 7.3. Sub-Lethal Exposures Induce Cellular Processes Potentially Leading to Tolerance

A study found that fifteen cycles of sub-lethal exposures of *S. aureus* to blue light (411 nm; 150 J/cm^2^ per cycle) resulted in the development of tolerance due to genetic alterations, which was stable after five successive sub-culturing. There was an increase in the expression of *recA* and *umuC* genes in the blue light-tolerant *S. aureus*, whereas mutant strains with non-functional *recA* and *umuC* genes did not develop tolerance. This finding confirmed that SOS-dependent mechanism played a role in the development of blue light-tolerant phenotype, although direct mutations from DNA damage were also possible [244]. In contrast, another study showed that *S. aureus* did not develop tolerance to blue light (405 nm; 108 J/cm^2^ per cycle) after fifteen cycles of sub-lethal treatment [246]. Exposures of *P. aeruginosa*, *A. baumannii* and *E. coli* to twenty cycles of blue light (405 nm; dosages enough to induce bacterial reduction of 4 log CFU) also did not result in the development of tolerance [251]. Possible explanations for the discrepancies between these studies include different wavelengths and blue light dosages used, albeit further investigations are needed to ascertain the effects of sub-lethal exposures of bacteria to blue light.

Two reports found that exposure of *S. aureus* to blue light had reduced its susceptibility to H_2_O_2_ [244,246]. In response to oxidative stress induced by the blue light, bacteria may up-regulate the expression of *katA* that encodes the production of H_2_O_2_-scavenging catalase protein and thus exhibit higher tolerance to H_2_O_2_ [118]. Adair and Drum identified thirty-two other genes in *S. aureus* regulated (up- or downregulated) by blue light (465 nm; 250 J/cm^2^), which included those responsible for the production of cell envelope components and heat-shock proteins [252]. Similarly, light-mediated gene regulations occurred in other major food-borne pathogens, namely *V. cholerae* and *C. sazakii*. In response to ROS generated by blue light (fluorescent black light; UV-filtered), *V. cholerae* showed differential expression of 222 genes (6.3%), relative to untreated cells, especially those encoding enzymes that protect or repair lipids and nucleic acids (genome)—these transcriptional responses to blue light were regulated by ChrR and MerR-like proteins [253]. Blue light (415 nm; up to 20.04 J/cm^2^) was also found to upregulate the expression of genes in *C. sakazakii* that encode oxidative stress-resistance chaperone, an adhesin and a capsule biosynthesis protein (CapC) [115].

These findings indicate that sub-lethal exposures of bacteria to blue light induce cell responses that may lead to the development of tolerance over time. However, complete resistance has not been reported, as even when tolerance was observed at a particular blue light dosage, increasing the light dosage was sufficient to eliminate the tolerant bacteria [244]. Nevertheless, other strategies are required to antagonize any development of bacterial tolerance to blue light. For example, the combination of blue light at 460 and 405 nm was reported to be effective against blue light-tolerant *S. aureus*. Leanse et al. [254] revealed that blue light at 460 nm (90–360 J/cm^2^) inactivated staphyloxanthin (a ROS scavenger; antioxidant) through photolysis and thus disrupted the ability of *S. aureus* to resist blue light treatment. Subsequent treatment with blue light at 405 nm (90–180 J/cm^2^) inactivated the bacteria (planktonic or in biofilm) at a higher rate than when single wavelengths were used, albeit inactivation was dependent on the dosage of both 405- and 460-nm blue light. In addition, as blue light attacks multiple targets in the bacterial cells, the development of resistance to blue light is likely to be slower than resistance to antibiotics.

## 8. Research Gap and Future Outlook

While it is a well-established fact that bacterial cells contain photo-active endogenous photosensitizers, the amount and type of these intracellular chromophores—and thus the susceptibility to blue light—may vary across bacterial species. For example, spectroscopic measurements revealed the presence of flavins and porphyrins in the cell lysates of *A. actinomycetecomitans*, although these compounds were not detected in *E. coli*. When illuminated by blue light (460 nm; 150 J/cm^2^), *A. actinomycetecomitans* (serotype b; ATCC 43718) were reduced by 5 log CFU, whereas *E. coli* (ATCC 25922) remained unaffected [255]. Another group of researchers also used spectroscopic technique to identify protoporphyrin IX and coproporphyrin as the main intracellular photosensitizers in *Helicobacter pylori*, which could exist in monomeric, dimeric or aggregated forms [256]. Others utilized high-performance liquid chromatography to characterize the endogenous photosensitizers in *P. aeruginosa* and *A. baumannii*, including coproporphyrin (I or III) and protporphyrin IX [257,258]. These studies provide technical foundation that future researches could build upon, particularly for characterizing endogenous photosensitizers in food-borne pathogenic bacteria.

There are limited data on the bactericidal activity of blue light against spoilage microorganisms, especially bacteria that are capable of growing in anaerobic conditions, such as *Clostridium estertheticum* in vacuum-packed meats. A study showed that while *E. coli*, *S. aureus* and *E. faecalis* in liquid media were unaffected by blue light (405 nm; 5.73 J/cm^2^) under anaerobic environments, the bacterial loads of *Prevotella intermedia* and *Prevotella nigrescenes* were significantly reduced by 1 and 2 log CFU/mL (*p* < 0.05), respectively [259]. Others observed 1-log reduction of *P. gingivalis* after illumination with blue light (405 nm; 3.42 J/cm^2^) [260]. The authors of both studies attributed blue light-mediated anaerobic inactivation of bacteria to the generation of organic radicals directly from the triplet state of endogenous photosensitizers [259,260]. These findings indicate that there is a potential for anaerobic application of blue light, although there is a need for improvements in the bactericidal efficacy of blue light under oxygen-scarce conditions. Azide salts can be used to facilitate anaerobic photodynamic treatments [216]. In addition, future studies are required to assess the effects of blue light on spore germination or the production of bacterial toxins.

Sensorial properties of blue light-treated foods need to be assessed beyond the quantitative measurements conducted within laboratory settings. For example, the quality of blue light-treated oysters were evaluated using both chemical/microbiological analyses and human panels (sensory evaluation), which provided the researchers with a comprehensive information to determine the shelf-life of seafood [212]. Future research can be designed to assess the sensorial properties of other types of blue light-treated food, including fruits, vegetables, meat and dairy products.

Lastly, spectral readings from Fourier-transform infrared (FTIR) indicated that blue light (470 nm) and UV (254 nm) primarily attacked the DNA during inactivation of MRSA, although the two lights targeted different DNA conformations: blue light induced damage on A-DNA, whereas UV predominantly inactivated B-DNA—these are two of the three conformations of double helical DNA, with the other one being Z-DNA. These findings suggest that blue and UV lights may be used as complementary treatments against microbes [261]. For safety, far ultraviolet-C (UV-C; 207 or 222 nm) can be used as an alternative to the conventional UV-C (254 nm). Accumulating evidence indicates that unlike the conventional UV-C, far UV-C exhibits bactericidal activity, but only has minimal effects on mammalian cells, such as the eye and skin of mice or human skin cells [262,263,264,265,266,267].

## Figures and Tables

**Figure 1 foods-09-01895-f001:**
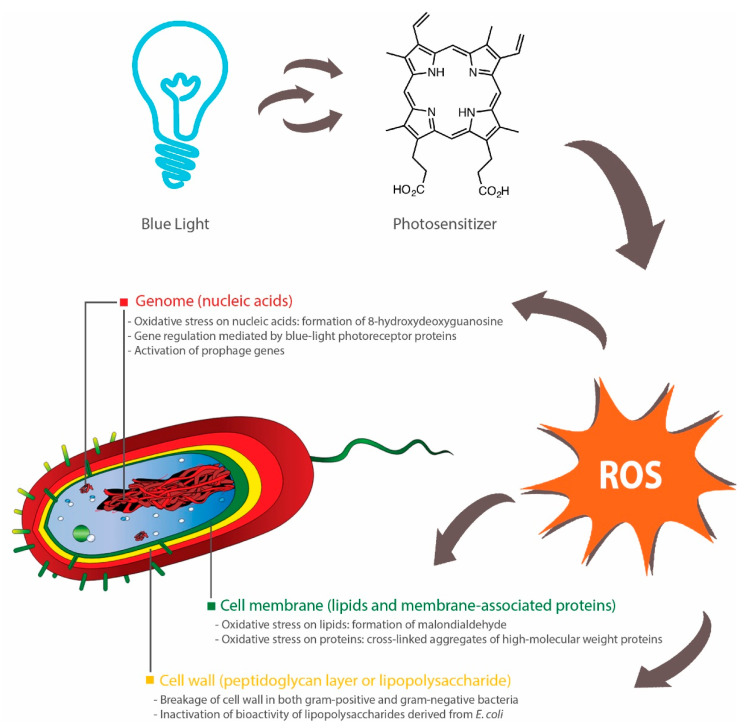
Bactericidal activities of blue light rely on activation of endogenous photosensitizers, such as porphyrins, which subsequently induces the production of reactive oxygen species (ROS). These ROS inflict oxidative damages to nucleic acids [116], lipids [113,115] and proteins [118]. Inhibition of biofilm formation can also occur through blue light-regulated transcriptional pathways [123,124,125] or bacterial inactivation through the activation of prophage genes [126]. Breakages of cell walls [120,121] and inactivation of lipopolysaccharides (outer membrane of Gram-negative bacteria) [122] have been reported, although the precise effects of antimicrobial blue light on peptidoglycan and lipopolysaccharide are not fully elucidated.

**Figure 2 foods-09-01895-f002:**
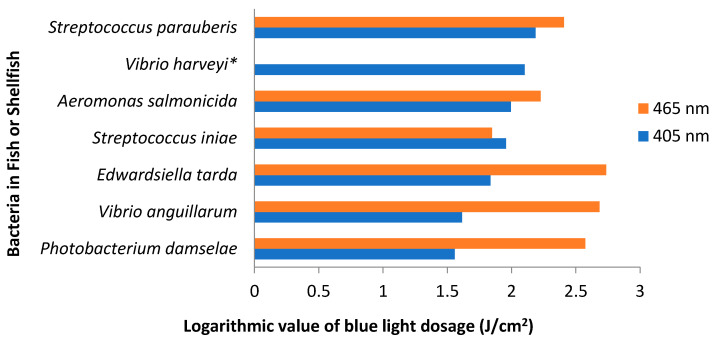
Blue light dosage required to achieve 1-log reduction of pathogenic bacteria in fish or shellfish. Light dosage was converted to logarithmic values (log) and thus an increase of one unit on the x axis represents a tenfold increase in the light dosage. This graph was created using data taken from Roh et al., which was published under the Creative Commons Attribution 4.0. International License (http://creativecommons.org/licenses/by/4.0/) [201]. * *Vibrio harveyi* was not inactivated by blue light at 465 nm.

**Table 1 foods-09-01895-t001:** Blue light inactivation of pathogenic bacteria on surfaces.

Bacteria	Surface	Reduction (CFU, CFU/mL or CFU/g)	Light Dosage	Light Source; Temperature; Distance; Photosensitizer ^β^	Reference
*B. cereus* (vegetative or spores)	Food packaging (yellow trays; LINPAC)	4 log (vegetative); 2.7 log (spores)	18 J/cm^2^	Blue LED (400 nm; 20 mW/cm^2^); ALA ^1^ (3 mmol/L)	[166]
*B. cereus* (vegetative or spores) *L. monocytogenes* (planktonic or biofilms)	Polyolefin food packaging (yellow trays; LINPAC)	4–4.5 log (vegetative); 2–5 log (spores) 4.2 log	3600–10,800 J (vegetative); 3600 J (spores) 3600 J	Blue LED (405 nm; 12 mW/cm^2^); CHL ^2^ (7.5 × 10^−7^–1.5 × 10^−6^ M for vegetative cells or 7.5 × 10^−6^–7.5 × 10^−5^ M for spores) Blue LED (405 nm; 12 mW/cm^2^); CHL ^2^ (7.5 × 10^−7^ M for planktonic cells or 1.5 × 10^−4^ M for biofilms)	[167]
*L. monocytogenes* (planktonic or biofilms)	Polyolefin food packaging (yellow trays; LINPAC)	2.3–3.7 log (planktonic); 1.7–3 log (biofilm)	18 J/cm^2^	Blue LED (400 nm; 20 mW/cm^2^); ALA ^1^ (7.5 or 10 mM)	[168]
Uropathogenic *E. coli*; *E. coli* O157:H7; *Salmonella* spp.; *L. monocytogenes*; *S. aureus*	STC ^2^ contaminated with bacteria-laden chicken purge	0.23–1.01 log	180 J/cm^2^	Blue LED (405 nm; 150 or 300 mW/cm^2^); 10 °C; 23 cm	[161]
*Campylobacter jejuni* *Campylobacter coli*	STC ^3^ contaminated with bacteria-laden chicken exudate	1.1 or 4.9 log 3.1 or 5.1 log	91.7 or 183.4 J/cm^2^ 89.2 or 185.8 J/cm^2^	Blue LED (405 nm; 151, 226 or 306 mW/cm^2^); 10 °C; 20.3 cm	[162]
*E. coli* *L. monocytogenes* *S. aureus* *P. aeruginosa*	Glass or acrylic surfaces	7–8 log (glass); 5 log (acrylic) ^α^ 2.48 (glass) 2.75 (glass) 3.72 (glass)	504 J/cm^2^ (E. coli) or 168 J/cm^2^ (other bacterial species)	Blue LED (405 nm; 141.48 mW/cm^2^); RT ^4^; 5 cm	[164]
*C. jejuni*	Stainless steel Cutting board (polyvinylchloride)	2 log 4 log	1.20–2.10 J/cm^2^	NUV–vis ^5^ LED (395 nm); RT ^4^; 3, 12 or 23 cm	[169]
*Salmonella enterica* subsp. *enterica* serovar Enteritidis *L. monocytogenes*	Polyvinylchloride (PVC) or acrylic surfaces	1.90–2.19 log (PVC); 1.18–1.63 log (acrylic) 0.68–0.90 log (PVC); 0.21–0.42 log (acrylic)	15–45 J/cm^2^ (PVC); 15–60 J/cm^2^ (acrylic)	Blue LED (405 nm; 110 mW/cm^2^)	[146]
*L. monocytogenes* (planktonic or biofilm)	STC ^3^ or AC ^6^ contaminated with bacteria-laden salmon exudate	Planktonic: 1.9–2.4 log (STC ^3^); 2.4–2.8 log (AC ^6^) Biofilm: 0.7–1.6 log	748.8 J/cm^2^	Blue LED (405 nm; 26 mW/cm^2^); 4, 15, 25 °C; 4.5 cm	[163]

^1^ ALA = 5-aminolevulinic acid (precursor in photosensitizer synthesis); ^2^ CHL = sodium chlorophyllin. ^3^ STC = stainless steel coupon; ^4^ RT = room temperature; ^5^ NUV–vis = near ultraviolet–visible (395 ± 5 nm); ^6^ AC = acrylic coupon. ^α^ Several concentrations of *E. coli* biofilm were tested (developed for 4 h to 72 h), but bacterial reductions presented in this review were only for those developed for 72 h (glass) or 48 h (acrylic). ^β^ Experimental temperature or distance were not specified in several studies, whereas photosensitizers were only used in some studies.

**Table 2 foods-09-01895-t002:** Blue light inactivation of pathogenic bacteria on dairy products.

Bacteria	Food Matrix	Reduction (CFU/mL or CFU/g)	Light Dosage	Light Source; Temperature; Distance ^β^	Reference
*E. coli*	UHT skim milk	4.69–5.27 log (405 nm); 4.11–5.04 log (433 nm); 3.41–4.64 log (460 nm)	Approx. 250 J/cm^2^ (405 nm); 313 J/cm^2^ (433 nm); 376 J/cm^2^ (460 nm) ^α^	Blue LED (405, 433 or 460 nm; 10 W); 5–15 °C; 30 mm	[174]
*S. aureus*; *E. coli*; *P. aeruginosa*; *S.* Typhimurium; *M. fortuitum*	Whole milk	5 log	228.84–583.5 J/cm^2^	Blue LED (413 nm; 100 mW/cm^2^); 1 mm	[171]
*P. fluorescens* (spoilage bacteria)	Ricotta cheese	3–5 log	6.36 J/cm^2^	Near UV–vis LED (395 nm; 16 mW/cm^2^); 6 cm	[176]
*L. monocytogenes * *P. fluorescens * (spoilage bacteria)	Sliced cheese (packaged)	5.14 log (4 °C); 1.95 log (25 °C) 3.60 log (4 °C); 1.85 log (25 °C)	604.8 J/cm^2^ (4 °C); 172.8 J/cm^2^ (25 °C)	Blue LED (460–470 nm; 1 mW/cm^2^); 4 or 25 °C; 10 mm	[165]
*S. enterica * (Gaminara, Montevideo, Newport, Typhimurium and Saintpaul)	Orange juice	2–5 log	4500 J/cm^2^	Blue LED (460 nm; 92, 147.7 or 254.7 mW/cm^2^); 4, 12 or 20 °C	[175]

^α^ Dosage for achieving maximum bacterial reduction was approximated using the formula: light intensity (10 W) × treatment lengths (s) at each wavelength/area of application (143.75 cm^2^). ^β^ Experimental temperature or distance were not specified in several studies.

**Table 3 foods-09-01895-t003:** Blue light inactivation of pathogenic bacteria in horticultural products.

Bacteria	Food Matrix	Reduction (CFU, CFU/mL or CFU/g)	Light Dosage	Light Source; Temperature; Distance; Photosensitizer ^β^	Reference
*L. monocytogenes *	Basil	1.6 log	9 J/cm^2^	Blue LED (405 nm; 10 mW/cm^2^); RT ^1^; 6 cm; chlorophyllin (1.5 × 10^−4^ M)	[186]
*E. coli *	Grape	2.4 log	36.3 J/cm^2^	Blue LED (465–470 nm; 4.5–30.2 mW/cm^2^); RT ^1^; 19 cm; curcumin (1.6 × 10^−3^ M)	[103]
*L. monocytogenes * *Salmonella * spp.	Cantaloupe rinds	At 405 nm: 2.4–2.9 log (no CHL); 2.8–3 log (CHL) At 460 nm: 2.7 log (no CHL); 2.2–2.3 log (CHL) At 405 nm: 2.3 (no CHL); 2.9 (CHL) At 460 nm: 1.1 log	1210 J/cm^2^ (405 nm); 5360 J (460 nm)	Blue LED (405 or 460 nm; 7 or 31 mW/cm^2^); 4 or 20 °C; CHL^2^ (100 µM)	[104]
*Salmonella * spp.	Fresh-cut papaya	1–1.2 log (4 °C); 0.3–1.3 log (10 °C); 0.8–1.6 log (20 °C)	900–1700 J/cm^2^	Blue LED (405 nm); 4, 10 or 20 °C; 2.3 or 4.5 cm	[116]
Mesophilic bacteria *B. cereus * *L. monocytogenes *	Cherry tomatoes	2.4 log 1.5 log 1.6 log	3–9 J/cm^2^	Blue LED (405 nm; 10 mW/cm^2^); RT ^1^; 6 cm NCCHL ^3^ (1.5 × 10^−4^ M)	[181]
*S.* Typhimurium	Strawberries	2.2. log	38 J/cm^2^	Blue LED (405 nm; 10–11 mW/cm^2^); 37 °C; 3.5 or 6 cm; CHL-CHN ^4^	[180]
*S.* Typhimurium	Cucumber peels	Approx. 3.9 log	18 J/cm^2^	Supra-luminous diode (SLD; 464 nm; 16.6 mW/cm^2^)	[187]
*E. coli * O157:H7 *E. coli * K-12 *S. Enteritidis* non-pathogenic *S.* Typhimurium	Almond kernel	1.43–2.44 log 1.64–1.84 log 0.55–0.70 log 0.64–0.96 log	2000 J ^§^	Blue LED (405 nm; 3.4 W); RT ^1^; 7 cm	[185]
*S. aureus *	Cucumber Pepper (green, red or yellow)	2.6 log 2.5 log	33.8 J/cm^2^	Blue LED (435 nm; 9.4 mW/cm^2^); RT ^1^; PVP-C ^5^ (50 or 100 µM)	[178]
*E. coli *	Cucumber Tomatoes Lettuce	3 log (10 µM); 4 log (50 µM); 4.5 log (100 µM) Approx. 3 log (10 µM); 6 log (50 µM); 3 log (100 µM) Approx. 3 log (10 µM); 7 log (50 µM); 6 log (100 µM)	33.8 J/cm^2^	Blue LED (435 nm; 9.4 mW/cm^2^);15 cm; cationic curcumin derivative (10, 50 or 100 µM)	[177]
*E. coli *	Fenugreek seeds Mung beans Mung bean germling	Approx. 3 log (10 µM); 5 log (50 µM); 4.5 log (100 µM) Approx. 2.5 log (10 µM); 2 log (50 µM); 3.5 log (100 µM) Approx. 0.5 log (10 µM); 1 log (50 µM); 0.5 log (100 µM)	33.8 J/cm^2^	Blue LED (435 nm; 9.4 mW/cm^2^);15 cm; cationic curcumin derivative (10, 50 or 100 µM)	[177]
*Salmonella * spp.	Fresh-cut pineapple	0.61–1.72 log	Approx. 8000 J/cm^2^	Blue LED (460 nm; 92–257 mW/cm^2^); 7, 16 or 25 °C; 2.5–4.5 cm	[184]
*E. coli * O157:H7, *Salmonella* spp. or *L. monocytogenes*	Fresh-cut mangoes	1–1.6 log	1700–3500 J/cm^2^	Blue LED (405 nm; 20 mW/cm^2^); 4, 10 or 20 °C; 4.5 cm	[182]
*L. monocytogenes * Mesophilic bacteria Yeasts and microfungi	Strawberries	1.8 log 1.7 log 0.87 log	14.4 J/cm^2^	Blue LED (400 nm; 12 mW/cm^2^); NCCHL ^3^ (1 mM)	[188]
*E. coli *	Fresh-cut Fuji apple	0.95 log	152 J/cm^2^	Blue LED (420 nm; 298 mW/cm^2^); 4 cm; curcumin (2 µM)	[183]

^1^ RT = room temperature; ^2^ CHL = sodium-copper chlorophyllin; ^3^ NCCHL = non-copperized sodium chlorophyllin; ^4^ CHL-CHN = non-copperized sodium chlorophyllin (1.5 × 10^−5^ M)-chitosan (0.1%). ^5^ PVP-C = Curcumin bound to polyvinylpyrrolidone. ^§^ Light dosage presented was used to treat three almond kernels (4 g) and approximated by multiplying the maximum treatment time (10 min) by light intensity (3.4 W). ^β^ Experimental temperature or distance were not specified in several studies, whereas photosensitizers were only used in some studies.

**Table 4 foods-09-01895-t004:** Blue light inactivation of pathogenic bacteria in meat and seafood products.

Bacteria	Food Matrix	Reduction (CFU, CFU/mL or CFU/g)	Light Dosage	Light Source; Temperature; Distance; Photosensitizer ^β^	Reference
Uropathogenic *E. coli*; *E. coli* O157:H7; *Salmonella* spp.; *L. monocytogenes*; *S. aureus*	Chicken skin	0.19–0.40 log	180 J/cm^2^	Blue LED (405 nm; 150 or 300 mW/cm^2^); 10 °C; 23 cm	[161]
*C. jejuni * *C. coli *	Chicken skin	1.7 log 2.1 log	184 J/cm^2^ 185.8 J/cm^2^	Blue LED (405 nm; 151, 226 or 306 mW/cm^2^); 10 °C; 20.3 cm	[162]
*L. monocytogenes *	Hot dog	<1 log	120 J/cm^2^	SLD (405 or 464 nm); 3–5 mm	[192]
*E. coli *	Hot dog	2.43 log	100 J/cm^2^	SLD (405 nm; 83.3 mW/cm^2^); 3–5 mm	[187]
*L. monocytogenes * *Salmonella * spp.	Fresh salmon	0.4 log (4 °C); 0.3 log (12 °C) 0.5 log (4 °C); 0.4 log (12 °C)	460.8 J/cm^2^	Blue LED (405 nm; 16 mW/cm^2^); 4 or 12 °C; 7.9 cm	[190]
*C. jejuni *	Skinless chicken fillet Chicken skin	1.43–2.62 log Approx. 6.7 log (3 cm); 1 log (12 cm); 0.7 log (23 cm)	1.20–2.10 J/cm^2^ 9 J/cm^2^ (3 cm); 4.23 J/cm^2^ (12 cm); 1.20 J/cm^2^ (23 cm)	NUV–vis ^1^ LED (395 nm); RT ^2^; 3, 12 or 23 cm	[169]
*L. monocytogenes *	Smoked salmon fillets	0.7–1.2 log	2400 J/cm^2^	Blue LED (460 nm; 15, 31 or 58 mW/cm^2^); 4 or 12 °C; 5.4–9 cm; riboflavin (25, 50 or 100 µM)	[191]
*S. aureus *	Chicken meat (with skin)	1.7 log	33.8 J/cm^2^	Blue LED (435 nm; 9.4 mW/cm^2^); RT ^2^; curcumin (50 or 100 µM)	[178]
*S. Enteritidis *	Cooked chicken	0.8–0.9 log	1.58–3.80 J/cm^2^	Blue LED (405 nm; 22 mW/cm^2^); 4 °C; 4 cm	[189]

^1^ NUV–vis = near ultraviolet–visible; ^2^ RT = room temperature. ^β^ Experimental distance was not specified in several studies, whereas photosensitizers were only used in some studies.

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
