# Peer review of "Antimicrobial Blue Light versus Pathogenic Bacteria: Mechanism, Application in the Food Industry, Hurdle Technologies and Potential Resistance"

_foods, 2020, doi:10.3390/foods9121895_

Round 1

Reviewer 1 Report

This is a manuscript about the effect of blue light on mainly pathogenic bacteria in food matrixes. Blue light as a decontamination procedure is rather interesting. In this manuscript an effort is being made to summarize most of the knowledge gathered towards the possible use of this technology to food chain decontamination. The manuscript is well written with an appropriate use of the English language. Still the report of the findings is made in a way that in several cases leaves some questions unanswered. Therefore the following points need to be edited.

Page 1, line 31 to page 2, line 61. Introduction. In this section, the authors are discussing why there is a need for blue light disinfection. Some information concerning on this technology is also needed.

Page 2, lines 50 – 55. The adverse effects of UVC should be reported in brief. In addition, the report of UVC adverse effects is made in such a way that it could be confused with those of other non-thermal food-processing technologies which is not true. Additionally, the possible disadvantages of other non-thermal food-processing technologies is also of interest (e.g. the cost or mechanical destruction of food by HPP).

Page 2, lines 69 – 74. In this paragraph, the contaminating, mostly commensal, bacteria are listed although the section is entitled as pathogenic bacteria. Their report is of great interest, and could be enhanced since this technology is of interest for decontamination of surfaces as well.

Page 4, line 157. In the mechanism section, a small paragraph could be added that summarizes the possible different mechanisms of action.

Page 7, line 275 and elsewhere. The name of E. coli and generally bacteria (Propionibacterium acne, e.t.c.) should be reported in italics

Page 7, line 319 and elsewhere. In vitro in italics

Page 9, lines 367 – 375. Although stated that this paragraph is about inactivation of Bacteria on Food Packaging and Working Surfaces, the authors are reporting the findings on liquids. Perhaps this is more suitable to the following paragraph.

Page 7, line 399 to page 11, line 443. I am wondering if the grouping of milk with other dairy products is suitable. I would expect that blue light has an efficacy on outer layers off dairy products such as cheese, whereas the liquid nature of milk is easier to decontaminate.

Page 12, lines 452 – 459. The inhibition of pathogenic bacteria with the combined use of other antimicrobial substances is rather interesting. As it is presented, no discrimination is made between the antimicrobial effect of the different substances per se.

Page 15, lines 556-565. I am wondering if this accurate. It is different to apply the blue light decontamination procedure for lowering the bacterial load of milk than treating the animal for infection, especially when the infection sight is rather difficult to reach.

Page 16, line 567 – 580. The bacteria reported are pathogenic mostly to fish. Since the manuscript is mostly about the inactivation of pathogenic to humans bacteria, a distinction should be made.

Page 17, lines 605 – 633. The introduction part was mainly about the food decontamination procedures that the blue light technology can substitute. Further, the shelf life extension is discussed in only this part. While reading this article, I would expect a rather detailed report on this. Since though a more detailed report is made on the following sections, perhaps this paragraph could act as an introductory section to the Hurdle Technology section.

Author Response

This is a manuscript about the effect of blue light on mainly pathogenic bacteria in food matrixes. Blue light as a decontamination procedure is rather interesting. In this manuscript an effort is being made to summarize most of the knowledge gathered towards the possible use of this technology to food chain decontamination. The manuscript is well written with an appropriate use of the English language. Still the report of the findings is made in a way that in several cases leaves some questions unanswered. Therefore the following points need to be edited.

The authors would like to thank the reviewer for their constructive comments, and the suggestions from the reviewer are addressed as follows:

Page 1, line 31 to page 2, line 61. Introduction. In this section, the authors are discussing why there is a need for blue light disinfection. Some information concerning on this technology is also needed.

We agree with the reviewer that there should be a clearer narrative on why blue light is an attractive decontamination technology. Thus, we added a sentence that highlights that blue light is safer than ultraviolet (Lin et al., 2013; REF 14), which allows for its wider application within the food supply chain (Lines 68-70).  

Page 2, lines 50 – 55. The adverse effects of UVC should be reported in brief. In addition, the report of UVC adverse effects is made in such a way that it could be confused with those of other non-thermal food-processing technologies which is not true. Additionally, the possible disadvantages of other non-thermal food-processing technologies is also of interest (e.g. the cost or mechanical destruction of food by HPP).

The description of harmful effects of UV-C was shortened (deleted Lines 64-66). To address the confusion mentioned by the reviewer, we moved the discussion on UV-C to a separate paragraph from the other non-thermal technologies. We added a review by D’Souza et al. (2015) [REF 8] to highlight the potential use of light as a cheap and sustainable decontamination technology (Lines 54-55).

Additionally, we have added a study by Sampedro et al. (2014) [REF 7], which provided data on the disadvantages of HPP and PEF, relative to thermal processing. These include higher consumption of electricity, and thus higher cost and carbon emission (Lines 49-53).   

Page 2, lines 69 – 74. In this paragraph, the contaminating, mostly commensal, bacteria are listed although the section is entitled as pathogenic bacteria. Their report is of great interest, and could be enhanced since this technology is of interest for decontamination of surfaces as well.

This is an insightful comment indeed. However, we found that the majority of studies on antimicrobial blue light focused on pathogenic and not on non-pathogenic bacteria, with only a few studies assessing spoilage bacteria (Table 2 and Section 5.3). Thus, we constructed the discussions in Section 2 to focus on pathogenic bacteria, as they are more relevant within the context of antimicrobial blue light. The mention of several commensal or other non-pathogenic bacteria is only intended for highlighting the susceptibility of food processing environments to bacterial contaminations.

Nevertheless, we agree with the reviewer that this is of great interest and can be considered a research gap that needs filling in the future. So, we expanded the discussion on Shelf Life section (Section 5.3.) by mentioning a finding of Moretro et al. (2017) [REF 16] who suggested that a range of spoilage bacteria could survive cleaning regimes (Lines 671-673). We also added several reviews articles on spoilage microorganisms of different foods and highlighted that these can be a subject of future studies (Lines 673-675).       

Page 4, line 157. In the mechanism section, a small paragraph could be added that summarizes the possible different mechanisms of action.

A summary was added at the end of the Mechanism section (Section 3.1.; Lines 243-250).

Page 7, line 275 and elsewhere. The name of E. coli and generally bacteria (Propionibacterium acne, e.t.c.) should be reported in italics

We would like to thank the reviewer for bringing this to our attention, as it seems that the journal de-italicized some of the scientific names during the formatting process, presumably by accident. Nevertheless, we have now changed them back to their original form, in which all scientific names are italicized.

Page 7, line 319 and elsewhere. In vitro in italics

Throughout the manuscript, in vitro is now italicized: Lines 275, 281, 329, 341, 347, 598, 720, 780 and 854.

Page 9, lines 367 – 375. Although stated that this paragraph is about inactivation of Bacteria on Food Packaging and Working Surfaces, the authors are reporting the findings on liquids. Perhaps this is more suitable to the following paragraph.

Well spotted, there is indeed a repetition (Lines 396-404 in Section 4.1. and Lines 450-455 in Section 4.2.) Thus, as the reviewer recommended, the discussion on liquids is not suitable in Section 4.1. and Lines 396-404 were deleted. Discussions on liquid inactivation are now provided in Lines 440-455 (Section 4.2.).

Page 7, line 399 to page 11, line 443. I am wondering if the grouping of milk with other dairy products is suitable. I would expect that blue light has an efficacy on outer layers off dairy products such as cheese, whereas the liquid nature of milk is easier to decontaminate.

We grouped the foods together based on the industry that they belong (dairy, animal-derived meats and plants), as they tend to have similar chemical compositions. The reviewer is correct that bacterial inactivation on solid cheeses – or any other solid foods – can only occur on the surface, which was why we incorporated orange juice into Section 4.2. as a comparison to liquid milk and had a separate paragraph for cheeses.

Throughout the manuscript, we have also presented evidence of how different application media could impact the bactericidal efficacy of blue light, for example, between smooth- and rough-surfaced vegetables (Lines 479-482) or peppers and PBS (Lines 484-487) or those with absorptive materials (Lines 522-526). Thus, readers can still easily understand that the efficacy of blue light is dependent on the application matrix. We also grouped foods according to their industry, such as putting liquid milk and cheeses together, so that potential readers from the commercial sector can systematically sort through the topics and find the most relevant ones based upon their area of interest.

Page 12, lines 452 – 459. The inhibition of pathogenic bacteria with the combined use of other antimicrobial substances is rather interesting. As it is presented, no discrimination is made between the antimicrobial effect of the different substances per se.

We added discussions on the antimicrobial mechanism of polyphenol (Lines 793-796) and essential oils (Lines 814-818), which primarily attack bacterial cell membrane. For nanoparticles, the mechanisms of silver nanoparticles and metal oxides are already provided in Section 6.2. (Lines 770-773 and Lines 779-782).

Page 15, lines 556-565. I am wondering if this accurate. It is different to apply the blue light decontamination procedure for lowering the bacterial load of milk than treating the animal for infection, especially when the infection sight is rather difficult to reach.

Yes, this is accurate. In this study, Moreira et al. (2018) firstly injected 1 mL of toluidine blue (2.5%) into the interior of the mammary gland by a catheter through the teat canal, then massaged the teat for better distribution of the photosensitizers, and subsequently lighted the teat using red LED. The LED was equipped with a cylindrical acrylic light guide, which was used to direct the red light into three different areas of the cow’s mammary glands. Lastly, milk samples were taken at the end of the treatment and analyzed. In the article, Moreira et al. also provided a schematic diagram and photos of how the treatment was done.

However, we share the reviewer’s concern about possible misinterpretations, albeit we still chose not to discuss this study in detail in our review to avoid disruption to the flow of the paragraph. Instead, we added a sentence that explains that the treatment was only meant to reduce incidences of mastitis in cows, using bacterial counts in milk as an indicator, but not as a direct decontamination method for milk (Lines 596-597).     

Page 16, line 567 – 580. The bacteria reported are pathogenic mostly to fish. Since the manuscript is mostly about the inactivation of pathogenic to humans bacteria, a distinction should be made.

We clarified this issue by explicitly stating that “…..most of these bacteria are not known to infect humans….” (Lines 608-609). We also added a sentence into the introduction to advise readers that some discussions on fish pathogenic bacteria concern non-human pathogens (Section 1; Lines 74-75).

Page 17, lines 605 – 633. The introduction part was mainly about the food decontamination procedures that the blue light technology can substitute. Further, the shelf life extension is discussed in only this part. While reading this article, I would expect a rather detailed report on this. Since though a more detailed report is made on the following sections, perhaps this paragraph could act as an introductory section to the Hurdle Technology section.

At the beginning, we did intend to construct detailed discussions on shelf-life and spoilage microorganisms. Unfortunately, to our dismay, there are limited data on these topics and these are the only studies that directly measured the shelf-life of foods treated with blue light.

The studies presented in the Hurdle Technology section primarily focused on pathogenic bacteria, and thus it may not be appropriate to link the Shelf Life section (Section 5.3.) and the Hurdle Technology section (Section 6). As per the reviewer’s recommendation in a previous comment, we did include additional review articles on the different spoilage microorganisms that could be targeted by blue light in future studies (Section 5.3. Lines 679-681).

Reviewer 2 Report

This manuscript describes current status of antimicrobial blue light against pathogenic bacteria, applicable to the food industry as an effective method for nonthermal processing. In this review, the authors also conclude limitation of relevant research and future prospect of blue light to control food pathogen with less harmful to human than UV. It is a meaningful review in that more demands for nonthermal food processing with pathogen control performance has been increasing recently. The contents and their sequence of this review are highly organized. Therefore, this review may be expected to provide concise and key features of blue light for pathogen control in foods. However, there are several comments that the authors should address and answer carefully as follows:

[Specific comments]

  1. Line 3: insert between “in” and “Food”
  2. In the entire manuscript, binominal scientific names of microorganisms, genes and species should be italicized. After the first appearance of the scientific name of an microorganism in the main text (from “Introduction” before “Author contribution”), the name of “genus” should be abbreviated.
  • Line 76 and other relevant lines: abbreviate “Staphylococcus” as “S.” (the first appearance in line 73)
  • Line 256, 257, 260, 261, 274, 275, and other relevant lines: Scientific names, genera and species of microorganisms should be italicized.
  1. Line 69, 70, and other relevant lines: change “gram-positive” and “gram-negative” with “Gram-positive” and “Gram-negative”
  2. Line 95: add “. (a period)” after “spp”
  3. Line 104: change “sp.” with “spp.”
  4. Line 126-128: Does this reference include any data of “Salmonella spp”, “Shigella spp.” and/or “Campylobacter perfringens”? Check the reference [75].
  5. Page 10, Table 1: correct “Enteridis” to “Enteritidis”
  6. Pagen 23-35, References: Check “Guide for Authors” to correct typographical errors, missing words, and formatting errors in the list.

Author Response

This manuscript describes current status of antimicrobial blue light against pathogenic bacteria, applicable to the food industry as an effective method for nonthermal processing. In this review, the authors also conclude limitation of relevant research and future prospect of blue light to control food pathogen with less harmful to human than UV. It is a meaningful review in that more demands for nonthermal food processing with pathogen control performance has been increasing recently. The contents and their sequence of this review are highly organized. Therefore, this review may be expected to provide concise and key features of blue light for pathogen control in foods. However, there are several comments that the authors should address and answer carefully as follows:

The authors wish to thank the reviewer for their constructive comments, and the suggestions from the reviewer are addressed as follows:

1. Line 3: insert between “in” and “Food”

We presume that the reviewer intended to suggest that the word “the” is inserted between “in” and “Food”, which we did.

2. In the entire manuscript, binominal scientific names of microorganisms, genes and species should be italicized. After the first appearance of the scientific name of an microorganism in the main text (from “Introduction” before “Author contribution”), the name of “genus” should be abbreviated.

  • Line 76 and other relevant lines: abbreviate “Staphylococcus” as “S.” (the first appearance in line 73)
  • Line 256, 257, 260, 261, 274, 275, and other relevant lines: Scientific names, genera and species of microorganisms should be italicized.

We would like to thank the reviewer for bringing this to our attention, as it seems that the journal de-italicized some of the scientific names during the formatting process, presumably by accident. Nevertheless, we have now changed them back to their original form, in which all scientific names, genera and species are italicized. We have also checked to ensure that abbreviation is used for the genus after its first appearance.

3. Line 69, 70, and other relevant lines: change “gram-positive” and “gram-negative” with “Gram-positive” and “Gram-negative”

All gram-positive and gram-negative are changed to Gram-positive and Gram-negative: those in lines 84, 86, 178-179, 215, 221 (in the description of Figure 1), 722-737, 841 and 847.

4. Line 95: add “. (a period)” after “spp”

A period was added after “spp” and now it is written “Clostridium spp.” (Line 109).

5. Line 104: change “sp.” with “spp.”

It was changed and now it is written “Salmonella spp.” (Line 117).

6. Line 126-128: Does this reference include any data of “Salmonella spp”, “Shigella spp.” and/or “Campylobacter perfringens”? Check the reference [75].

The authors thank the reviewer for being vigilant. The reviewer is correct that the mentioned study (now it is REF 78) only detected L. monocytogenes and E. coli, and not the other bacteria. We also re-checked subsequent references (REF 79-84) and separately listed the names of bacteria isolated from the different food products (Lines 141-144).

7. Page 10, Table 1: correct “Enteridis” to “Enteritidis”

It was corrected and now it is written as “Salmonella Enteritidis”.

8. Pagen 23-35, References: Check “Guide for Authors” to correct typographical errors, missing words, and formatting errors in the list.

We corrected the referencing style, including the in-text citations.